# Gentirigeoside B from *Gentiana rigescens* Franch Prolongs Yeast Lifespan via Inhibition of TORC1/Sch9/Rim15/Msn Signaling Pathway and Modification of Oxidative Stress and Autophagy

**DOI:** 10.3390/antiox11122373

**Published:** 2022-11-30

**Authors:** Lan Xiang, Dejene Disasa, Yanan Liu, Rui Fujii, Mengya Yang, Enchan Wu, Akira Matsuura, Jianhua Qi

**Affiliations:** 1College of Pharmaceutical Science, Zhejiang University, 866 Yu Hang Road, Hangzhou 310058, China; 2Department of Biology, Graduate School of Science, Chiba University, Chiba 263-8522, Japan

**Keywords:** *G. rigescens* Franch, Gentirigeoside B, TORC1 signaling pathway, autophagy, oxidative stress, longevity

## Abstract

Gentirigeoside B (GTS B) is a dammaren-type triterpenoid glycoside isolated from *G. rigescens* Franch, a traditional Chinese medicinal plant. In the present study, the evaluation of the anti-aging effect and action mechanism analysis for this compound were conducted. GTS B significantly extended the replicative lifespan and chronological lifespan of yeast at doses of 1, 3 and 10 μM. Furthermore, the inhibition of Sch9 and activity increase of Rim15, Msn2 proteins which located downstream of TORC1 signaling pathway were observed after treatment with GTS B. Additionally, autophagy of yeast was increased. In addition, GTS B significantly improved survival rate of yeast under oxidative stress conditions as well as reduced the levels of ROS and MDA. It also increased the gene expression and enzymatic activities of key anti-oxidative enzymes such as Sod1, Sod2, Cat and Gpx. However, this molecule failed to extend the lifespan of yeast mutants such as ∆*cat*, ∆*gpx*, ∆*sod1*, ∆*sod2*, ∆*skn7* and ∆*uth1*. These results suggested that GTS B exerts an anti-aging effect via inhibition of the TORC1/Sch9/Rim15/Msn signaling pathway and enhancement of autophagy. Therefore, GTS B may be a promising candidate molecule to develop leading compounds for the treatment of aging and age-related disorders.

## 1. Introduction

Aging is a complex and progressive biological process that leads to vulnerability to a variety of chronic diseases such as cardiovascular and neurodegenerative disorders as well as metabolic syndromes [1]. As the number in the aging population increases worldwide, the burden of aging and age-related impact on socio-economic status is enormous [1,2,3]. Even though several promising anti-aging active molecules have been identified, there are no FDA-proved commercially available drugs to delay aging. Therefore, studies for the discovery of small molecules which could enhance quality of life and elongate lifespan and thereby act as a protective measure to alleviate age-related disorders and furnish a healthy lifespan are highly desirable.

The target of rapamycin (TOR) is a nutrient-sensing protein kinase which is evolutionarily conserved in eukaryotic organisms [4]. Evidence indicates that the TOR signaling pathway is a central regulator that connects nutrient availability and stress condition to control cell growth and lifespan [4,5,6]. The inhibition of the TORC1 signaling pathway can delay aging and extend the lifespan of different species such as *Saccharomyces cerevisiae*, *Caenorhabditis elegans*, *Drosophila melanogaster* and mice [6,7,8,9]. In the TORC1 signaling pathway, Sch9, a serine-threonine protein kinase, is a downstream protein of TORC1 positively regulated by TORC1. Inhibition of Sch9 can protect cells from oxidative stress conditions and extend replicative and chronological lifespans of yeast [10,11]. Protein kinase Rim15 and transcription factors Msn2/4 are other downstream elements in the TORC1/Sch9 signal transduction. Rim15 and Msn2/4, which are negatively controlled by the TORC1 signaling pathway, regulate important anti-oxidative genes, such as *SOD1* and *SOD2*, which are essential for lifespan extension through reduction of reactive oxygen species (ROS). Gradual accumulation of ROS causes oxidative stress, which is implicated to play a crucial role in the aging process. These excessive ROS are scavenged by anti-oxidative enzymes such as Sod, Cat and Gpx [12]. Furthermore, the initiation of autophagy is directly affected by TORC1 activity.

Silent information regulator 2 (Sir2), an NAD^+^-dependent deacetylase protein, controls replicative lifespan in yeast [5]. Activation of Sir2 plays an indispensable role in yeast lifespan extension via inhibition of extrachromosomal rDNA circle formations. Homologous recombination of repetitive rDNA is the primary cause of aging in *S. cerevisiae*. These repetitive combinations of rDNA are stabilized under normal conditions by Sir2 [4]. Several studies reported that the TORC1 signaling pathway controls the activity of Sir2 in regulating the lifespan of *S. cerevisiae* [4,5,12].

Budding yeast, *Saccaromyces serevisiae*, is one of the known model organisms in anti-aging research [13]. Among available strains, K6001, which is derived from W303, has a characteristic that only mother cells in glucose medium can produce daughter cells, making it suitable to evaluate the anti-aging effect via replicative lifespan assay. The replicative lifespan is generally defined as the number of progenies produced by the cell division of a single yeast cell until death. Short generation time, low cost and relatively simple operations are the advantages of using K6001 as a model organism [13,14]. In our previous studies, we reported several promising anti-aging molecules using the K6001 bioassay system for screening [14,15,16,17,18]. Therefore, this method was also used in the current study to screen for anti-aging small molecules from the Chinese medicinal herb, *Gentiana rigescens* Franch (Jian Long Dan in Chinese).

*Gentiana rigescens* Franch is one of the well-known traditional Chinese medicinal plants. Traditionally, the root of this plant is used to treat different ailments [19]. In our previous studies, 11 novel neuritogenic benzoate-type molecules (gentisides) from this plant were discovered. Moreover, the NGF-mimic effects of these molecules and action of mechanism were clarified in PC12 cells [20]. Meanwhile, the memory improving effect using mixtures of gentisides was evaluated in an Alzheimer’s disease mouse model induced by scopolamine. We found that these molecules and derivatives exerted anti-AD function via inhibition of acetylcholine activity, oxidative stress and regulation of insulin-like growth factor 1 receptor/extracellular signal-regulated kinase pathways [21,22]. In addition, anti-aging gentipicroside and amarogentine, secoiridoid glycosides-type anti-aging active molecules, were isolated and mechanisms of action of these molecules were investigated in our previous studies [14,18]. In the present study, we identified another anti-aging dammaren-type triterpenoid glycoside, gentirigeoside B (GTS B), from this plant with the replicative lifespan assay of K6001 yeast and gained further insight into the mechanism of action of GTS B. Here, we reported that GTS B extended the lifespan of yeast via inhibition of the TORC1/Sch9/Rim15/Msn signaling pathway and the regulation of oxidant stress and autophagy.

## 2. Materials and Methods

### 2.1. General

Silica gel (200–300 mesh, Yantai Chemical Industry Research Institute, Yantai, China) and reversed-phase C_18_ open columns (Cosmosil 75 C_18_ OPN, Nacalai Tesque, Kyoto, Japan) were used for column chromatography. Pre-coated silica gel (0.25 mm) and RP-18 plates (0.25 mm) (Yantai Jiangyou Silicone Gel Development Co., Ltd., Yantai, China) were used for TLC analysis. Preparative high-performance liquid chromatography (HPLC, Eilite, Dalian, China) was conducted, equipped with two Elite-230 pumps, and a UV detector was used for purification. High-resolution electrospray ionization mass spectrometry (HR-MS) analysis was performed on Agilent 6224A accurate mass Time-of-Flight LC/MS system (Agilent Technologies, Santa, CA, USA). A Bruker AV III-500 spectrometer (Bruker, Billerica, MA, USA) was employed for NMR measurement. The ^13^C NMR chemical shifts in *δ* (ppm) referred to solvent chemical shifts of *δ*_C_ (150.350) for pyridine-*d*_5_.

### 2.2. Preparation of Gentirigeoside B from G. rigescens Franch

Dried roots of *G. rigescens* were purchased from Huqingyutang Pharmacy, Hangzhou, Zhejiang Province, China. The plant was identified by associate Professor Jianxia Mo and the voucher specimen (no. 20190620) was deposited in the Institute of Material Medica, Zhejiang University. The dried root of the medicine (483.36 g) was powdered and extracted with 80% aqueous methanol for 48 h under shaking at room temperature. The extract obtained was partitioned between ethyl acetate and water. Both fractions were tested with the replicative lifespan assay of K6001 yeast. The active water layer was subjected to ODS open column using a methanol/water mixture as a mobile phase. The active fraction obtained from 50/50 and 70/30 methanol/water mobile phase was subjected to ODS open column under a CH_3_OH/H_2_O solvent system for further separation. Finally, the active fraction obtained from 45/55 CH_3_OH/H_2_O was purified by HPLC [SP ODS-A (20 × 250 mm)] using 55% aqueous methanol and a flow rate of 8 mL/min to obtain an active sample (4.4 mg, 17 min), which was identified as gentirigeoside B through comparison of the spectroscopic results with those previously reported in the literature [23]. ^13^C NMR (125 Hz, pyridine-*d*_5_): *δ* = 178.7, 105.1, 104.6, 90.8, 82.4, 81.3, 79.7, 79.0, 78.4, 78.8, 78.7, 77.5, 75.9, 71.9, 71.1, 70.7, 63.7, 62.8, 61.7, 56.7, 51.0, 50.2, 45.8, 44.8, 43.9, 40.6, 39.2, 36.7, 35.8, 33.2, 31.8, 28.1, 27.8, 27.2, 26.9, 26.0, 22.7, 22.1, 18.6, 15.5, 16.4 and 16.3 (Appendix A). HRESI-TF-MS *m/z* showed a molecular mass of 869.4479, calculated for C_42_H_70_O_17_Na (M+Na)^+^ 869.4480. The chemical structure of this compound is shown in Figure 1a.

### 2.3. Yeast Strain, Culture Medium and Lifespan Assay

K6001 with background W303, BY4741, BY4741-sf-GFP-SCH9-5HA, BY4741-RIM15-GFP and BY4741-MSN2-GFP yeast strains, the mutants of *uth1*, *skn7*, *sod1*, *sod2*, *cat* and *gpx* of K6001 yeast, and YOM36 and YOM38- GFP-ATG8 yeast strains were used in this experiment. The used mediums were composed of 1% yeast extract, 2% peptone and 2% D-glucose (YPD) or 1% yeast extract, 2% peptone and 3% galactose (YPG). The replicative lifespan experiment was performed based on a previous procedure [14]. Briefly, the K6001 yeast strain was cultured in 5 mL YPG medium for 24 h at 28 °C under consistent 180 rpm shaking. Afterwards, 1 mL of broth containing yeast was washed with PBS three times and the obtained pellet was diluted with PBS to count the cells using a haemocytometer. Approximately 4000 cells were spread on YPD agar plates containing resveratrol (RES) or GTS B at different concentrations. Afterward, the agar plates were incubated for 48 h at 28 °C. Forty microcolonies formed on each agar plate were randomly selected to count the number of daughter cells produced by one mother cell. The replicative lifespan of mutants of *uth1*, *skn7*, *sod1*, *sod2*, *cat* and *gpx* with the K6001 background were performed the same as that of the K6001 yeast strain. Chronological lifespan refers to the survival time of yeast cells during undivided and stable periods. It was assessed as described in a previous report [14]. Briefly, YOM36 was incubated in synthetic complete medium containing 2% glucose, 2% peptone and 1% yeast extract or SD medium (0.17% yeast nitrogen base without amino acid or ammonium sulfate, 0.5% ammonium sulfate and 0.2% glucose) in a shaking incubator at 28 °C. After incubation for 24 h, the cells were inoculated in SD medium containing GTS B at 0, 1, 3 and 10 µM with a yeast initial OD value of 0.01 and cultured for 72 h at 28 °C. Afterwards, roughly 200 yeast cells of each group were scribbled on glucose agar plates and incubated for 48 h. Colony-forming units were counted from each plate and the colonies on the third day were defaulted to one hundred percent survival. This process was repeated every two days and the survival rate was calculated from the third day (colony-forming unit of each plate of every two days/colony-forming unit of the third day for each group × 100%). The process was terminated after the survival rate became less than 10%, which was designated as one hundred percent death. The genome types of yeast strains are described in Appendix A.

### 2.4. Real-Time Polymerase Chain Reaction (RT-PCR) Analysis

Firstly, the BY4741 yeast strain was cultured with GTS B at doses of 0, 1, 3 and 10 µM or RES at a dose of 10 µM in YPD liquid medium under 180 rpm shaking at 28 °C for 24 h. Consequently, total RNA was extracted by the hot-phenol method as reported previously [16]. cDNA was synthesized by the reverse transcription method using 5 µg of the extracted RNA and HiFi-MMLV cDNA kits (Cowin Biotech, Beijing, China). Finally, quantitative RT-PCR was performed using CFX96 Touch (Bio-Rad, Hercules, FL, USA) and SYBR premix Ex Taq (Takara, Otsu, Japan). The sequence of primers used in this analysis is described in Appendix A. After normalizing the data to TUB1 levels, 2^−ΔΔCt^ formula was used to estimate the relatively transcribed mRNA levels. Average values were considered after running the samples in triplicate.

### 2.5. Nuclear Translocation of Rim15 and Msn2 Proteins Analysis

To check whether GTS B affected the TORC1 signaling pathway, we constructed RIM15-GFP and MSN2-GFP yeast strains with the BY4741 background, which are located downstream of TOCR1, to observe the activity or nuclear translocation of these proteins. Initially, these yeast strains stored at −30 °C were inoculated on YPD agar plates. After forming colonies on plates, the colonies of these yeasts were inoculated in YPD liquid medium and cultured in a shaker at 180 rpm and 28 °C for 10 h. Subsequently, the MSN2-GFP yeasts at 0.1 initial OD_600_ in each group were treated with GTS B at 0, 1, 3 and 10 μM and rapamycin at 1 μM as positive control for 2 h. Meanwhile, the yeast of RIM15-GFP firstly was cultured for 3 days and was subsequently treated with GTS B at different concentrations for 6 h. These yeasts of each group were washed with PBS and stained with Hoeschst 33342 (final concentration, 1 μg/mL) for 7 min in the dark. Thereafter, the cells were cleaned with PBS to remove remaining Hoeschst 33342 from the background. Finally, the nuclear translocations of Msn2-GFP and Rim15-GFP were observed using a two-photon confocal fluorescence microscope (Olympus FV1000BX-51, Tokyo, Japan) or fluorescence microscope, respectively.

### 2.6. T-Sod, Sod1 and Cat Enzymes Activity Assay

BY4741 yeast cells were cultured for 12 h. Thereafter, the cultivated cells at the same OD value were treated for 24 h with RES 10 µM or GTS B at 0, 1, 3 and 10 µM and divided into five groups for each concentration. Subsequently, the cells were ultrasonicated on ice for 5 min and the cell lysates obtained were centrifuged at 12 × 10^3^ rpm at 4 °C for 15 min to collect the supernatant for the activity test. Cat, T-Sod and Sod1 activities were tested using the Cat Assay Kit (Beyotime Biotechnology Limited Company, Shanghai, China) and Sod Assay Kit (Nanjing Jiancheng Biotechnology Institute, Nanjing, China).

### 2.7. Yeast Growth Experiment under Oxidative Stress Condition

BY4741 yeast strain with an initial OD_600_ value of 0.01 was treated with GTS B at 0, 1, 3 and 10 µM or RES 10 µM and incubated for 24 h. Thereafter, 5 µL of cultured yeast with the same OD_600_ value from different groups was dropped on YPD agar plates containing 9 mM H_2_O_2_. Optimum concentration of H_2_O_2_ was selected based on our previous studies [15,16,17]. The growth of yeast cells from each group was observed and photographed after incubation for three days. For quantitative purposes, we used the same yeast strain but a different method. BY4741 was treated with RES at 10 µM or GTS B at 0, 1, 3 and 10 µM. Approximately 200 yeast cells of each group were smeared on a glucose agar plate with or without 5 mM H_2_O_2_. After two days, grown yeast cells were counted for each group. The survival rate of yeast cells was analyzed from the ratio of the number of colonies with H_2_O_2_ divided by the number of colonies without H_2_O_2_.

### 2.8. Measurement of MDA, ROS Levels of Yeast

The ROS level of yeast was measured using the method reported [14]. Briefly, the BY4741 yeast strain was treated with RES 10 µM or GTS B at 0, 1, 3 and 10 µM for 24 h. Then, 2′, 7′-dichloro-dihydroflourescein diacetate (DCFH-DA) was added to the cultured cell to obtain the final concentration of 10 µM. The cells were then incubated in the dark under a shaker for 1 h. Subsequently, the cells were harvested and washed with PBS three times. DCF fluorescence intensity of approximately 1 × 10^7^ cells was detected using a SpctraMax M3 multimode fluorescence plate reader (Molecular Devices Corporation, San Francisco, CA, USA) under 488 nm of excitation and a 525 nm emission wavelength. MDA levels of yeast were measured following a previously reported method [15]. The cells were cultured in the same way as for ROS determination; they were cleaned with PBS three times then suspended in 500 µL PBS and ultrasonicated on ice for 5 min. Subsequently, they were centrifuged at 12 × 10^3^ rpm for 15 min at 4 °C to collect the supernatant. Finally, MDA level was measured using the MDA Assay Kit (Nanjing Jiancheng Bioengineering Institute, Nanjing, China).

### 2.9. Visualization of Autophagy Induction in Yeast

The experiment was conducted as previously described [14]. The YOM38 yeast strain containing pR316-GFP-ATG8 plasmid was incubated in the dark with YPD medium and shaking at 180 rpm. After 24 h, cells were cleaned with PBS, divided into different groups with the same OD_600_ value and treated with GTS B at different concentrations of 0, 1, 3 and 10 µM and RES at 300 μM as positive control. Cells were cultured for 22 h, washed with PBS and stained with DAPI (4′, 6 diamidino-2-phenylindol 20 μg/mL) for 10 min in dark, and then the cells were cleaned with PBS to remove remaining DAPI from the background. Finally, induction of autophagy of yeast was observed using a two-photon confocal fluorescence microscope (Olympus FV1000BX-51, Tokyo, Japan).

### 2.10. Western Blotting Analysis

At first, the sfGFP-SCH9-5HA yeast strain was spread on YPD agar plates at −30 °C. After forming colonies on plates, one of the yeast colonies was inoculated in YPD liquid medium and cultured in a shaker at 180 rpm at 28 °C for 12 h. Subsequently, the yeast cultures were divided into four groups at the same initial OD_600_ value and treated with GTS B at 0, 1, 3 and 10 μM and RES at 300 μM for 2 h. The protein samples were prepared with sonicated yeast cells in RAPI lysis buffer, centrifugation and measured protein concentration. Afterward, 20 μg of yeast proteins was separated by sodium dodecyl sulfate polyacrylamide gel electrophoresis and transferred to polyvinylidene fluoride membranes (Bio-Rad Laboratones, Inc., Hercules, FL, USA). The membranes were then incubated with primary antibodies specific to HA (16B12, BioLegend, San Diego, CA, USA) and β-actin (#CW0096, CoWin Biotech, Beijing, China). The secondary antibody, horseradish peroxidase-linked goat anti-mouse IgGs (#CW0102, CoWin Biotech, Beijing, China), was used for HA and β-actin. Protein bands were visualized by using the e-ECL Western Blot Kit (CoWin Biotech, Beijing, China) and analyzed via ImageJ software (National Institute of Health, Rockville, MD, USA).

### 2.11. Statistical Analysis

All data were presented as mean ± SEM and each experiment was repeated at least three times. Bio-statistical analysis was performed with one-way ANOVA followed by Tukey’s post hoc test using GraphPad Prism software v9.4.1 (GraphPad Prism, San Diego, CA, USA). The value of *p* less 0.05 was considered as a statistically significant difference.

## 3. Results

### 3.1. GTS B Prolongs Replicative and Chronological Lifespan of Yeast Cells

Yeast is a prominent model organism to study aging mechanisms and identify potential anti-aging drug candidates [24]. In this particular study, the K6001 replicative lifespan bioassay system was used to identify anti-aging active molecules. RES was used as a positive control. The anti-aging potential of GTS B was evaluated at doses of 1, 3 and 10 µM. In the replicative lifespan assay, the average lifespan of each group was as follows: 7.60 ± 0.56 generations in the control group; 10.35 ± 0.68 (*p* < 0.01) in the RES-treated group at 10 µM; and 9.70 ± 0.71 (*p* < 0.01), 10.83 ± 0.72 (*p* < 0.001) and 10.05 ± 0.71 (*p* < 0.01) generations in the GTS B-treated groups at 1, 3, and 10 µM, respectively, as shown in Figure 1b. Furthermore, the chorological lifespan of YOM36 yeast was used to confirm the antiaging effect of GTS B. The survival rates in the 3 and 10 μM GTS B treatment groups were also significantly increased as shown in Figure 1c. These results suggest that GTS B is a potential anti-aging molecule.

### 3.2. Effect of GTS B on the Gene Expression of TORC1, RPS26A, RPL9A and SIR2

The TOR signaling pathway is a central regulator that connects nutrient availability and stress condition to control cell growth and lifespan. Inhibition of the TORC1 signaling pathway delays aging and extends lifespan by manipulating the activities of downstream kinase proteins, transcription factors and NAD^+^-dependent deacetylase enzyme Sir2 in *Saccharomyces cerevisiae* [5,6,12]. Inhibition of TORC1 signaling reduces expression of ribosomal protein genes such as RPS26A and RPL9A, indicating downregulation of protein synthesis and upregulation of the anti-aging process [12,25]. Based on this hypothesis, the effects of GTS B on the gene expression of key proteins such as TOCR1, Rps26a*,* Rpl9a and Sir2 in TOR signaling pathways were investigated and results are shown in Figure 2a–d. The relative gene expressions of TORC1, RPS26A and RPL9A in treatment groups were not significantly changed as compared to the control group after treatment of GTS B for 24 and 48 h (Figure 2a–c). Therefore, we examined whether GTS B affects the gene expression of SIR2 and results are shown in Figure 2d. The abundance of SIR2 mRNA in the 1 μM GTS B-treated groups and the RES group at 24 h was significantly increased as compared to the control group. However, the gene expression of SIR2 in GTS B treatment groups at 48 h was not obviously changed in comparison with the control group. These results suggest that GTS B does not affect the TOR signaling pathway at the gene level, but it can regulate gene expression of the Sir2 protein.

**Figure 1 antioxidants-11-02373-f001:**
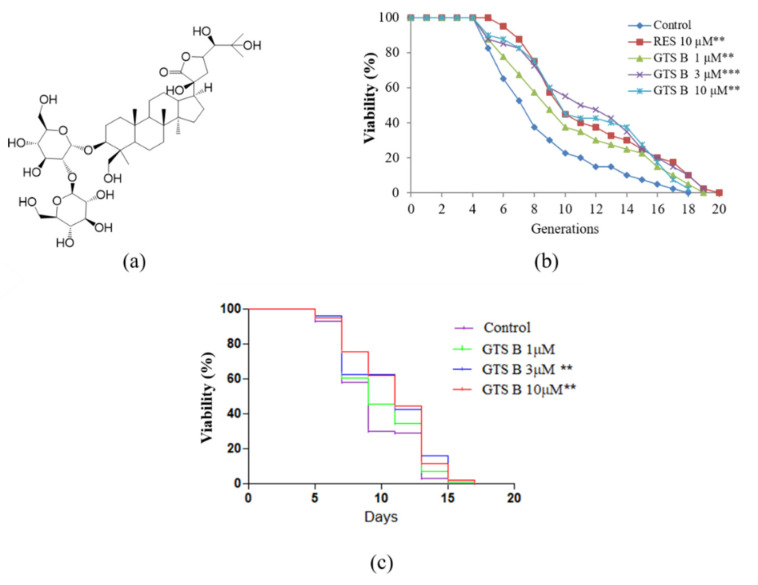
Chemical structure of GTS B and its effect on replicative and chronological lifespan of K6001 and YOM36 yeast strains, respectively. (**a**) Chemical structure of GTS B, (**b**) Replicative lifespan of K6001 yeast cells and (**c**) Chronological lifespan of YOM36 yeast cell after treatment with 0, 1, 3 and 10 µM GTS B or RES (positive control) at a dose of 10 µM. The experiment was repeated three times. ** and *** represent a significant difference as compared to untreated group for *p <* 0.01 and 0.001, respectively.

### 3.3. Effect of GTS B on the Gene Expression and Protein Level of Sch9

Sch9 is a substrate and major effector of the highly conserved TOCR1, a key protein which controls cell growth in response to different nutrients and stresses [8]. Thus, the gene expression and protein level of Sch9 were measured after giving GTS B. The abundance of SCH9 mRNA in the 10 μM GTS B-treated group at 24 h was obviously changed in comparison with the control group (Figure 3a, *p* < 0.05). There were no changes in SCH9 gene expression in other GTS B treatment groups at 24 h and all groups at 48 h (Figure 3a). Furthermore, we observed changes in the Sch9 protein level of BY4741-sfGFP-Sch9-5HA yeast with the Western blot analysis and anti-HA antibody. As we expected, the protein levels of Sch9 were significantly decreased by GTS B and RES (Figure 3b and Appendix A
*p* < 0.001, *p* < 0.001, *p* < 0.001, *p* < 0.001). These results clarified that GTS B exerted the anti-aging effect via inhibition of the TORC1 signaling pathway.

### 3.4. Effect of GTS B on the Gene Expression and Activity of Rim15

In addition, Rim15 is a key integrator of signals from a variety of lifespan-regulating pathways. It is also a known activator of stress resistance transcription factors, such as Mns2/4, which are stress resistance transcription factors responsible for the regulation of important anti-oxidative genes [11]. We also detected the changes in gene expression for these genes after treatment with GTS B for 24 and 48 h (Figure 4a). The significant increase in RIM15 gene expression was only observed in the 10 μM GTS B-treated group for 24 h (Figure 4a, *p* < 0.05). Therefore, we constructed the BY4741-GFP-RIM15 strain to check the nuclear location of Rim15 after treatment with GTS B for 6 h. The nuclear transfer location of Rim15 in BY4741-GFP-RIM15 yeast cells was significantly increased after treating with GTS B at 1, 3 and 10 μM and RES at 300 μM (Figure 4b,c, *p* < 0.05, *p* < 0.05, *p* < 0.01, *p* < 0.01). These results suggest that Rim15 is involved in the anti-aging effect of GTS B.

### 3.5. Effect of GTS B on the Gene Expression of MSN2 and MSN4 and Activity of Msn2

The stress transcription factors Msn2/4 are located downstream of Rim15. To check if GTS B also affects Msn2/4 at mRNA and protein levels, we also detected the gene expression and nuclear translocation of Msn2/4. The gene expression of MSN2 was significantly increased in the 10 μM GTS B-treated group for 24 h and 1 μM GTS B-treated group for 48 h (Figure 5a, *p* < 0.05, *p* < 0.05). However, the abundance of MSN4 messenger RNA was not affected by GTS B at all doses for 24 and 48 h (Figure 5b). Furthermore, we detected the nuclear translocation of Msn2 after treating GTS B. The nuclear translocation of Msn2 was significantly increased by GTS B at doses of 3 and 10 μM and RES at 300 μM (Figure 5c,d, *p* < 0.001, *p* < 0.01, *p* < 0.05). These results suggest that Msn2 takes part in the anti-aging effect of GTS B.

### 3.6. GTS B Improves Gene Expression of SOD1, SOD2 and CAT

Sod1, Sod2 and Cat are endogenous anti-oxidants that act to suppress or prevent free radicals or reactive oxygen species. The enzymes of which these genes transcribed are keys to forestall aging through prevention of the damaging effect caused by ROS [23]. Gene expression of SOD1, SOD2 and CAT were significantly increased after treatment with 1, 3 and 10 μM of GTS B and RES at 10 μM (Figure 6a–c, *p* < 0.01, *p* < 0.001, *p* < 0.01, *p* < 0.05, *p* < 0.01, *p* < 0.05). Furthermore, Sod and Cat are first-line anti-oxidant key enzymes. They suppress or prevent formation of reactive species or free radicals in cells. In addition, they convert toxic substances into harmless products, thereby preventing oxidative stress [26]. Therefore, the effects of GTS B upon the activity of these enzymes were evaluated at different concentrations. As indicated in Figure 6d–f, the enzyme activity was significantly increased after treatment with RES at 10 μM and GTS B at doses of 3 and 10 µM (*p* < 0.01, *p* < 0.001, *p* < 0.001; *p* < 0.001, *p* < 0.01, *p* < 0.05; *p* < 0.01, *p* < 0.01, *p* < 0.01). These results indicated that GTS B generated an anti-aging effect via improving the activities of key anti-oxidative enzymes.

### 3.7. Effects of GTS B on Oxidative Stress of Yeast

Oxidative stress is an underlying factor for aging and age-related disorders. ROS serves as a signaling molecule for physiological responses, causing oxidative modifications to macromolecule-like proteins, nucleic acids and lipids at excessive levels. Such oxidative damages ultimately lead to aging and age-related diseases [27,28]. Therefore, the anti-oxidative potential of GTS B was assessed to highlight its potential role as an anti-aging molecule. The quantitative and qualitative results of this experiment are shown in Figure 7a,b. GTS B at all doses improved the growth of yeast under oxidative stress induced by 9 mM H_2_O_2_ (Figure 7a). Furthermore, the survival rates of treated groups were increased significantly as compared to the untreated control group in quantitative experiments under oxidative stress induced by 5.5 mM H_2_O_2_ (Figure 7b, *p* < 0.001, *p* < 0.001, *p* < 0.001).

In addition, exposure of lipid macromolecules to reactive free radicals leads to a series of chain reactions which result in lipidperoxidation. MDA is a known toxic lipidperoxidation product which can cause cellular and tissue damage. It is one of the typical biomarkers of oxidative stress [29,30]. ROS plays an indispensable role in metabolism. When production of ROS surpasses the desired limit due to compromised eliminations or excessive productions, the subsequent imbalance results in oxidative stress [30]. Therefore, the effect of GTS B on the MDA and ROS levels of yeast were investigated. As it is indicated in Figure 7c, the levels of MDA in yeast were significantly decreased in the GTS B-treated groups as compared to the control group (*p* < 0.001, *p* < 0.001 and *p* < 0.001). Similarly, a significant decrease in ROS of yeast in GTS B treatment groups is also observed in Figure 7d (*p* < 0.001, *p* < 0.001 and *p* < 0.001). These results confirmed that GTS B exhibited beneficial anti-aging effects through protection from oxidative stress.

### 3.8. GTS B Failed to Prolong the Replicative Lifespan of Mutants of K6001 Yeast-Related Oxidative Stress

Anti-oxidant enzymes are a frontline defense system against oxidative stress. Sod is a major anti-oxidant playing an indispensable role in yeast survival. Yeast lifespan could be prolonged by overexpression of SOD genes [31]. Cat helps cells to survive by breaking down reactive hydrogen peroxide into products such as water and oxygen. It is used as therapeutic agent for several diseases related to oxidative stress. The cellular Gpx system is a part of the essential constituents protecting cells against oxidative stress by detoxifying hydrogen peroxide in cells. Gpx plays a crucial role in protecting cells from oxidative damage exerted by free radicals, especially lipid peroxidation. The aging gene, UTH1, regulates oxidative stress and its deletion can lead to prolonged lifespan [32]. Skn7 is a transcriptional factor that responds to oxidative stress signals [33]. Therefore, the effect of GTS B on the replicative lifespan of oxidative stress-related mutants was evaluated. As indicated in Figure 8, the replicative lifespan of K6001 is 7.60 ± 0.56 generations for the control group, 10.35 ± 0.68 generations for the RES-treated group and 10.83 ± 0.72 generations for the GTS B-treated group (3 µM). Figure 8a shows the *sod1* mutant with replicative lifespans of 7.50 ± 0. 47 generations for the control group, 8.83 ± 0.61 generations for the 10 µM RES group and 7.85 ± 0.59 generations for the 3 µM GTS B group. The replicative life span of the *sod2* mutant in Figure 8b revealed 7.25 ± 0.57, 7.18 ± 0.60 and 7.63 ± 0.71 generations for control, RES- and GTS B-treated groups, respectively. The replicative lifespans of the *cat* mutant in Figure 8c are 7.83 ± 0.61, 9.50 ± 0.71 and 9.08 ± 0.63 generations for control, RES- and GTS B-treated groups, respectively. The replicative lifespans of the *gpx* mutant in Figure 8d are 8.68 ± 0.64, 9.18 ± 0.78 and 9.20 ± 0.68 generations for control, RES- and GTS B-treated groups, respectively. Figure 8e indicates the replicative lifespans of the *skn7* mutant yeast in control, 10 µM RES and 3 µM GTS B-treated groups are 7.43 ± 0.61, 7.38 ± 0.64 and 7.50 ± 0.64, respectively. The replicative lifespan of the *uth1* mutant of K6001 yeast has increased from 7.48 ± 0.06 to 10.18 ± 0.095 compared with the K6001 background. After giving 10 µM RES and GTS B at a dose of 3 μM, the replicative lifespan in the RES group and GTS B group are 10.55 ± 0.67 and 11.58 ± 0.72 generations, respectively (Figure 8f). Taken together, we found no significant difference between treated and control groups for all oxidative stress-related genes of mutant yeast cells, suggesting that these anti-oxidative enzymes are involved in GTS B-related anti-aging effects.

### 3.9. Effect of GTS B on Autophagic Induction in Yeast

Autophagy is a degradation pathway that eliminates aggregated protein and damaged cellular organelles implicated to cause aging and age-related disease. The process of autophagy improves the quality of cytoplasm via eradication of distorted cellular macromolecules and dysfunctional organelles. The targeting of autophagy for the discovery of novel therapeutics is highly desirable as it regulates aging and age-related pathologies [1,2,25]. Therefore, we speculated that the GTS B molecule might induce autophagy in yeast cells. In this experiment, YOM38-GFP-ATG8 yeast was used to investigate the changes in autophagy after treatment with the GTS B molecule. As it is shown in Figure 9a,b, autophagic flux was induced significantly in all treated groups as compared to the control group *(p* < 0.001, *p* < 0.01, *p* < 0.001, *p* < 0.001). These results clarified that the GTS B molecule can modulate the process of autophagy to improve yeast lifespan.

## 4. Discussion

*G. rigescens* Franch is one of the prominent traditional Chinese medicinal plants [19]. Previously, our research group isolated several potential novel neuritogenic and anti-aging molecules from this plant. For some of the isolated molecules, structures were modified semi-synthetically for structure-activity relationship studies, and a promising lead compound was identified and developed through this study [14,18,20,21,22,23]. For our intensive study on this plant, we isolated an active dammaren-type glycosidic triterpenoid, GTS B, using K6001 yeast cells as a bioassay method. The results of replicative lifespan and chronological lifespan in Figure 1 indicate that GTS B has anti-aging effects on yeast.

The TOR signaling pathway plays an important role in regulation of the lifespan of yeast. It is a Ser/Thr protein kinase, which was firstly isolated in *Saccharomyces cerevisiae*, and consists of rapamycin-sensitive TOR complex 1 (TORC1) and rapamycin-insensitive TOR complex 2 (TORC2) [34,35]. The core of TORC1 includes TOR, lethal with SEC13 protein 8 (LST8) and a regulatory-associated protein of TOR. It controls cell proliferation and temporal growth via keeping the balance between anabolic and catabolic processes [36,37]. TORC2, which mainly contains TOR, LST8, stress-activated map kinase-interacting protein 1 and rapamycin-insensitive companion of TOR, regulates spatial cell growth by modulating the cytoskeleton structure, cell polarity, glycolysis, glycogenesis, lipogenesis and gluconeogenesis [34,37,38]. Pharmacologic inhibition or deletion of TORC1 can delay aging and extend lifespan across different species [4,5,6,7,8,9]. In this study, to gain further insight into the mechanism of action of GTS B, we detected several gene expressions, protein level of Sch9 and nuclear translocation of some proteins in TOR signaling pathways. The gene expressions of TOCR1, RPL9A, RPS26A, RIM15 and MSN4 were not affected by GTS B in Figure 2, Figure 4 and Figure 5. Therefore, we constructed the BY4741-sf-GFP-SCH9-5HA, BY4741 -RIM15-GFP and BY4741- MSN2-GFP yeast strains to detect the activity change of these proteins. The reduction in Sch9 protein and nuclear translocation of Rim15-GFP and Mns2-GFP in Figure 3, Figure 4 and Figure 5 demonstrate that the TOR signaling pathway is involved in the anti-aging effects of GTS B.

Msn2/4 are stress resistance transcription factors responsible for the regulation of important anti-oxidative genes. To understand whether the Sod and Cat enzymes were directly regulated by Msn2/4 transcription factors, we focused on further elucidation of gene expression and enzyme activity for selected enzymes related to anti-oxidative effects. The significant increase in gene expressions of SOD1, SOD2 and CAT and activity of these enzymes in Figure 6 clarified that Msn2 affected SOD and CAT gene expression and increased the activity for these enzymes. Furthermore, we measured the changes in survival rate of yeast under oxidative stress and biomarkers such as ROS and MDA in normal conditions after giving GTS B. The results in Figure 7 revealed that GTS B decreased oxidative stress by increasing anti-oxidative gene expression and anti-oxidative enzymatic activity. To obtain further elucidation as to which of these proteins are involved in the anti-aging effects of GTS B, we constructed mutants of *sod1*, *sod2*, *gpx*, *cat*, *skn7* and *uth1* yeast with the K6001 background and performed a replicative lifespan assay. No changes in replicative lifespan of these mutants are observed in Figure 8, indicating that these proteins play an important role in the anti-aging effect of GTS B.

The activity of TORC1 directly affects the initiation of autophagy. When nutrition is sufficient, activated TORC1 can phosphorylate Atg13, thereby inhibiting the formation of the Atg1/Ulk1 complex to inhibit autophagy. The decapping enzyme DCP2 phosphorylated by TORC1 is acidified to bind to the RNA helicase RCK, thereby affecting the transcription of ATG genes and inhibiting autophagy [39,40]. Therefore, we also detected the changes in autophagy of yeast after giving GTS B. The increase in autophagy of yeast in Figure 9 revealed that GTS B induced autophagy to play a role in anti-aging via the TOR signaling pathway.

Interestingly, we found that GTS B significantly increased the gene expression of SIR2, which is another longevity gene and exists in interaction with TORC1 (Figure 2d). It gives us an indication that the Sir2 signaling pathway is one of the key research directions for us in the future.

In the present study, bioassays guided the isolation to obtain GTS B as an anti-aging compound from an extract of *Gentiana rigescens* Franch, and content of GTS B in the plant is very low (0.0009% in dried weight). Due to the small amount of GTS B on hand and also the instability of the molecule in acidic condition, we did not try acid hydrolysis to obtain the aglycone. We tried to obtain the aglycone of GTS B from the less-polar fraction. However, we did not isolate the aglycone till now. Therefore, we did not know whether the presence of the sugar moiety was necessary or not necessary for the activity. Additionally, it is not easy to obtain enough GTS B for animal experiments in a short time. In the future, we will focus on these points to conduct intensive study.

In summary, GTS B from *G. rigescens* Franch has shown a significant anti-aging effect on budding yeast. GTS B prolongs yeast lifespan through inhibition of the TORC1/Sch9/Rim15/Msn2 signaling pathway and modification of autophagy and oxidative stress (Figure 10). This study lays a foundation for deep biological action mechanisms and chemical studies such as structural modifications of GTS B in anti-aging and age-related drug discovery research.

## Figures and Tables

**Figure 2 antioxidants-11-02373-f002:**
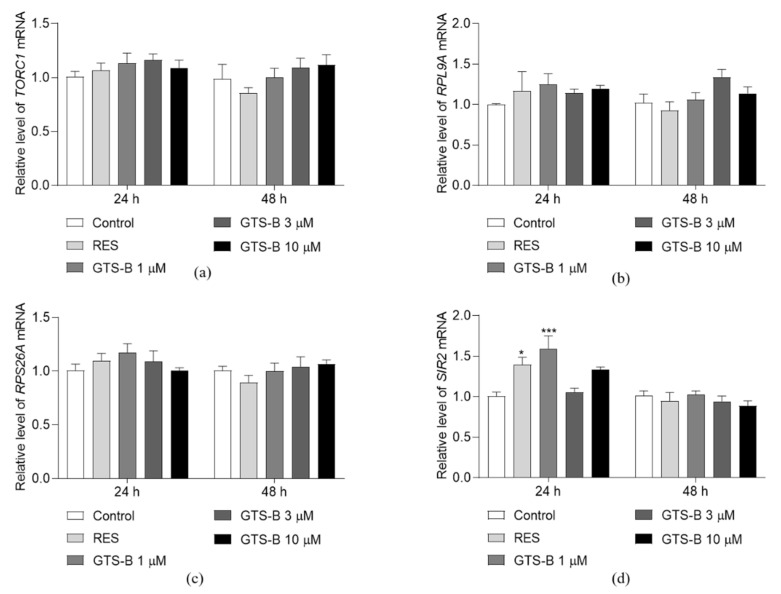
Effect of GTS B on the gene expression of TORC1 (**a**), RPL9A (**b**), RPS26A (**c**) and SIR2 (**d**). BY4741 yeast cells were cultured with GTS B at 0, 1, 3 and 10 µM, and 10 μM RES for 24 and 48 h. * and *** represent a significant difference at *p* < 0.05 and 0.001 as compared to control group, respectively.

**Figure 3 antioxidants-11-02373-f003:**
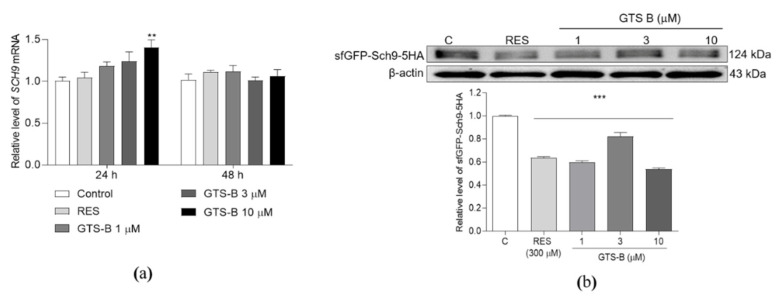
Effect of GTS B on the gene expression and protein level of Sch9. (**a**) The changes in gene expression of SCH9 after treatment with GTS B for 24 and 48 h. (**b**) The changes in Sch9 protein levels of BY4741 after treatment with GTS B at doses of 1, 3 and 10 μM. ** and *** represent significant differences as compared to control group at *p* < 0.01 and 0.001, respectively.

**Figure 4 antioxidants-11-02373-f004:**
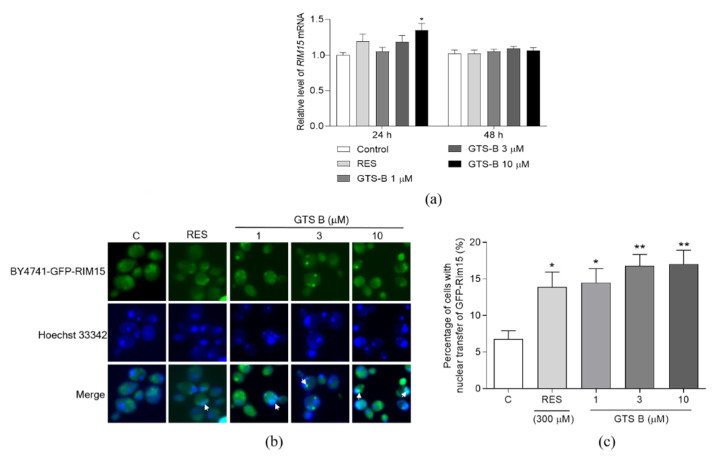
Effect of GTS B on the gene expression of RIM15 and nuclear translocation of Rim15 in yeast. (**a**) The changes in gene expression of RIM15 after treatment with GTS B for 24 and 48 h. (**b**) The photograph of nuclear translocation of Rim15 after treatment with GTS B for 6 h. (**c**) The digital results of nuclear translocation of Rim15 after treatment with GTS B for 6 h (**b**). Each value is the average of five repetitions of each group. * and ** indicate a significant difference between control and treated groups at *p* < 0.05 and *p* < 0.01.

**Figure 5 antioxidants-11-02373-f005:**
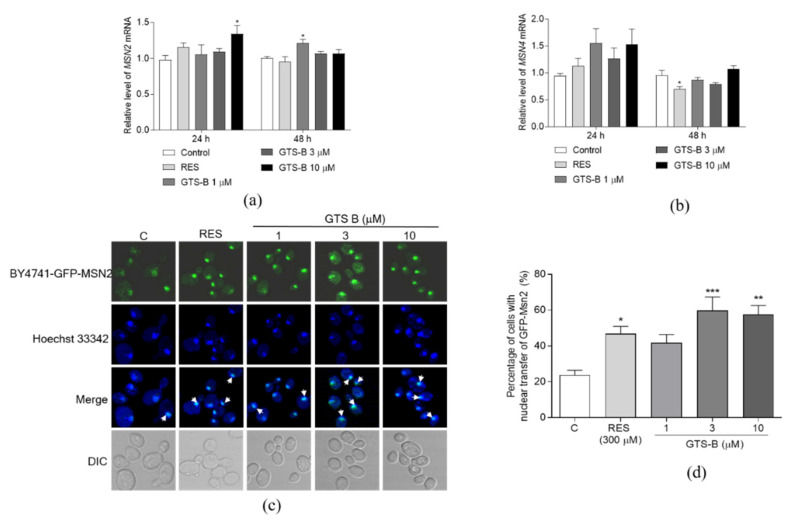
Effect of GTS B on the gene expression of MSN2, MSN4 and nuclear translocation of Msn2. (**a**,**b**) The changes in gene expression of MSN2 and MSN4 after treatment of GTS B for 24 and 48 h. (**c**) The photograph of nuclear translocation of Msn2 after giving GTS B for 2 h. (**d**) The digital results of nuclear translocation of Msn2 after giving GTS B for 2 h (**c**). Each value is the average of five repetitions of each group. *, ** and *** represent a significant difference as compared to control group at *p* < 0.05, *p* < 0.01 and *p* < 0.001, respectively.

**Figure 6 antioxidants-11-02373-f006:**
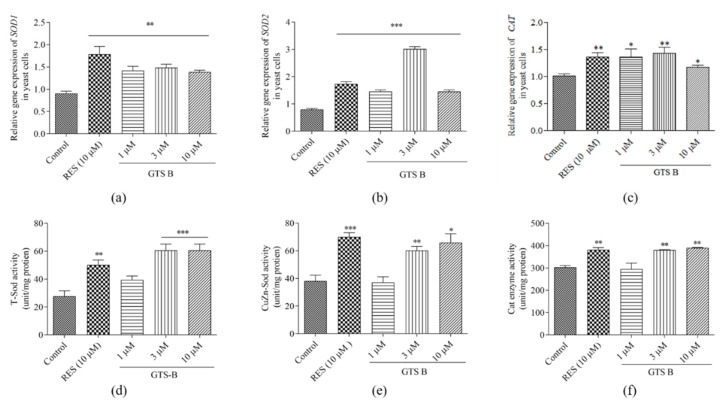
Effect of GTS B on the gene expression and enzyme activity of Sod1, Sod2 and Cat. (**a**–**c**) The changes in gene expression of SOD1, SOD2 and CAT. (**d**–**f**) The changes in enzyme activity of Sod1, Sod2 and Cat. Each value is the average of five repetitions of each group. *, ** and *** show a significant difference as compared to control group at *p* < 0.05, *p* < 0.01 and *p* < 0.001, respectively.

**Figure 7 antioxidants-11-02373-f007:**
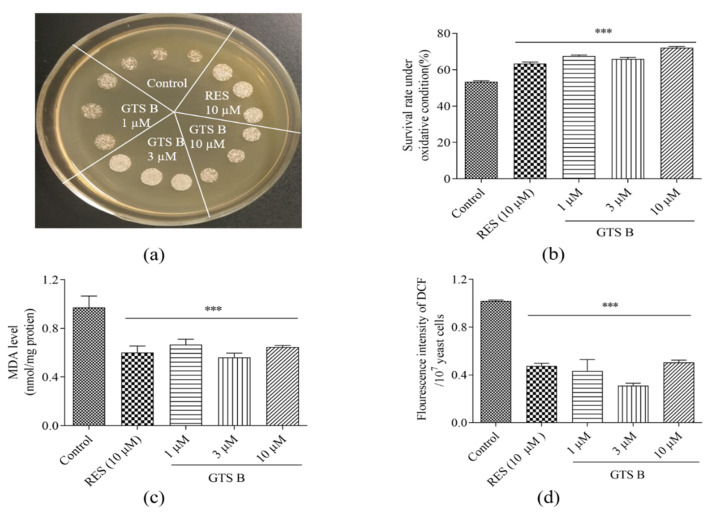
Effect of GTS B on the growth of BY4741 yeast cells under oxidative stress, ROS and MDA of yeast. (**a**) Photograph of yeast growth under oxidative stress induced by 9 mM H_2_O_2_ after treatment with GTS B of different concentrations. (**b**) Quantitative description of BY4741 yeast survival rate under oxidative stress induced by 5 mM H_2_O_2_. (**c**) The changes in MDA level of yeast after culturing with GTS B at different concentrations. (**d**) The changes in ROS level in yeast after giving GTS B or RES at different concentrations. Each value represented as mean of each group ± SEM and repeated numbers were five. *** indicates a significant difference as compared to control groups at *p* < 0.001.

**Figure 8 antioxidants-11-02373-f008:**
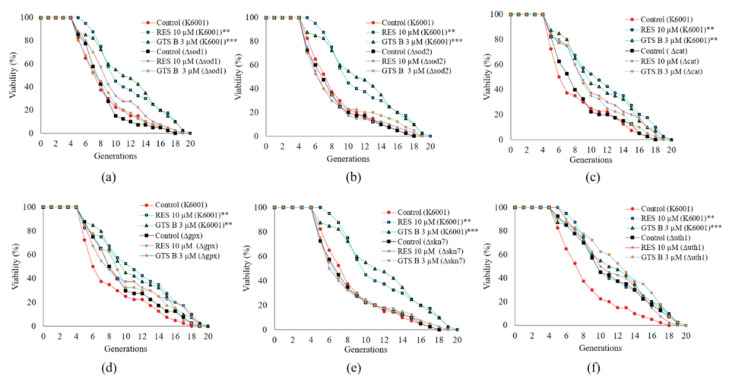
Effect of GTS B on the replicative lifespans of yeast mutants with K6001 background. The replicative lifespan changes in (**a**) ∆*sod1*, (**b**) ∆*sod2*, (**c**) ∆*cat*, (**d**) ∆*gpx*, (**e**) ∆*skn7* and (**f**) ∆*uth1* yeasts with K6001 background. The yeast K6001 and its mutants were treated with GTS B at 3 µM or RES at 10 µM and cultured for 48 h at 28 °C. Forty randomly selected colonies were used to count daughter cells produced by one mother cell. The experiment was repeated three times. ** and *** express significant differences as compared with untreated group at *p* < 0.01 and *p* < 0.001, respectively.

**Figure 9 antioxidants-11-02373-f009:**
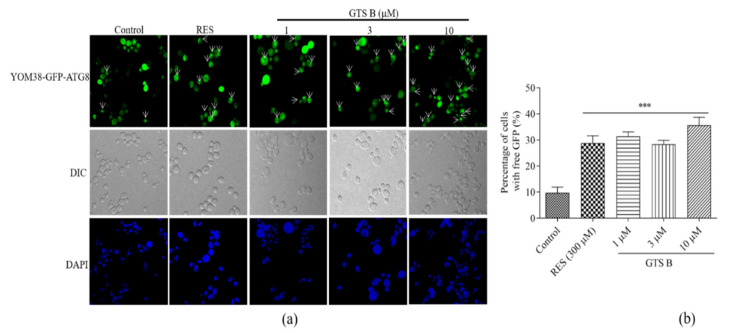
Effect of GTS B on autophagy induction in YOM38-GFP-ATG8 yeast. (**a**) The photograph of autophagy induced by GTS B was observed under confocal fluorescence microscope. (**b**) The digital result of induced autophagy in yeast after treatment of GTS B. Each value is the average of cells with GFP in six visual fields. *** indicates a significant difference at *p* < 0.001.

**Figure 10 antioxidants-11-02373-f010:**
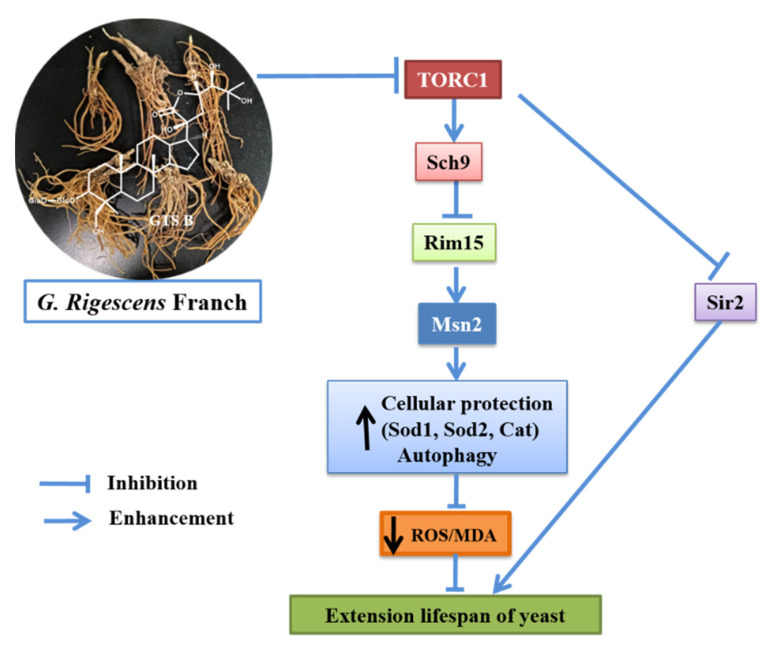
Proposed mechanism of action for anti-aging GTS B.

## Data Availability

All figures and data used to support this study are included within this article and Appendix A.

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
