# Peer review of "Gentirigeoside B from Gentiana rigescens Franch Prolongs Yeast Lifespan via Inhibition of TORC1/Sch9/Rim15/Msn Signaling Pathway and Modification of Oxidative Stress and Autophagy"

_antioxidants, 2022, doi:10.3390/antiox11122373_

Round 1

Reviewer 1 Report

Xiang and colleagues demonstrate the effects of gentirigeoside B from G. rigescens on the prolongation of yeast lifespan via inhibition of of TORC1/Sch9/Rim15/Msn Signaling 3 Pathway and Modification of Anti-Oxidative Stress and AutophagyThe manuscript depicts very interesting findings, but some minor issues should be addressed.

1. As a glycosidic derivative, its metabolic stability should be determined. Is the presence of the sugar moiety necessary for activity? The observed activity is also maintained in the aglycone form or only in the glycoside form?

2. line 16: at doses of 1, ....10 M. is it M or mM or uM?

3. In the title author should provide the full botanical name of the plant.

Author Response

Reply for reviewer 1

Referee 1

( ) English language and style

( ) English very difficult to understand/incomprehensible
( ) Extensive editing of English language and style required
( ) Moderate English changes required
(x) English language and style are fine/minor spell check required
( ) I don't feel qualified to judge about the English language and style

Yes

Can be improved

Must be improved

Not applicable

Does the introduction provide sufficient background and include all relevant references?

(x)

( )

( )

( )

Are all the cited references relevant to the research?

(x)

( )

( )

( )

Is the research design appropriate?

(x)

( )

( )

( )

Are the methods adequately described?

(x)

( )

( )

( )

Are the results clearly presented?

(x)

( )

( )

( )

Are the conclusions supported by the results?

(x)

( )

( )

( )

Comments and Suggestions for Authors

Xiang and colleagues demonstrate the effects of gentirigeoside B from G. rigescens on the prolongation of yeast lifespan via inhibition of of TORC1/Sch9/Rim15/Msn Signaling 3 Pathway and Modification of Anti-Oxidative Stress and Autophagy. The manuscript depicts very interesting findings, but some minor issues should be addressed.

Question 1. As a glycosidic derivative, its metabolic stability should be determined. Is the presence of the sugar moiety necessary for activity? The observed activity is also maintained in the aglycone form or only in the glycoside form?

Answer: Thank you very much for your good suggestion. In the present study, bioassay guided the isolation to obtain GTS B as an anti-aging compound from extract of Gentiana rigescens Franch and content of GTS B in the plant is very low (0.0009% in dried weight). Due to the small amount of GTS B in hand and unstable of the molecule in acidic condition, we did not try acid hydrolysis to obtain the aglycone. We tried to obtain the aglycone of GTS B from less polar fraction. However, we did not isolate the aglycone till now. Therefore, we did not know the presence of the sugar moiety is necessary or not necessary for the activity. Besides, it is not easy to obtain enough amount of GTS B for animal experiment in the short time. In the future, we will focus on these points to do intensive study.

In addition, we added this information in P14, L505-512 of discussion section

Question 2. line 16: at doses of 1, ....10 M. is it M or mM or uM?

Answer: It is mM and we revised it of P1, L16 as following:

GTS B significantly extended the replicative lifespan and chronological lifespan of yeast at doses of 1, 3 and 10 mM.

Question 3. In the title author should provide the full botanical name of the plant.

Answer: We accepted your comment and revised the title of manuscript as following:

Gentirigeoside B from Gentiana rigescens Franch Prolongs Yeast Lifespan via Inhibition of TORC1/Sch9/Rim15/Msn Signaling Pathway and Modification of Oxidative Stress and Autophagy

Reviewer 2 Report

The authors are presenting their research regarding Gentirigeoside B from G. rigescens and the effect of the compound in yeast related to Modification of lifespan, anti-oxidative stress and autophagy. The article is well presented, however I have some critical points that I feel needs to be corrected before this manuscript can be accepted for publication.

Between different species, processes, molecular pathways, and homologous genes are not always the same. This problem is not often discussed, especially within pharma applications, and I feel that we need to address this matter more. It is good that animal models are not used in the current study, since animal models are out-dated and should to a larger extent be replaced by more modern way to study pathways and efficacy. I also think that using yeast as a model organism is good since it enables a rapid, cost efficient and sustainable way further explore pathways and mechanisms. However, I would also like to see a small section where a discussion regarding the use of drawbacks when using model organisms, such as yeast that is used in the current study, to solve a problem on another organism, human. I would like the authors to address this topic and to which degree human cell models could have been/can be applied instead.

Below are my specific comments and suggestions for improvement.

Throughout the manuscript, please correct the line breaks so that they do not occur in the middle of words. For abbreviations, always write out the full name the initial time that the abbreviation is used, e.g. for BSA at row 147.

Row 14, I suggest to write “In the present study evaluation…” instead of “In the present study. The evaluation…”.

Row 18-19, I suggest to write “were observed after treatment with GTS B. Also, autophagy of yeast ….” if one aspiration timepoint was used or “were observed after treatment of GTS B. At the same 18 time, autophagy of yeast …..”  

Row 20, I suggest to write “….and reduced the levels of ROS and MDA.” or similar instead of “and reduced ROS and MDA levels in yeast, respectively.”.

Row 21, I suggest to write “It also increased the gene expression and enzymatic…...” or similar instead of “Meanwhile, it increases the genes expression and enzymatic….”.

Row 31-31, I suggest to write “Aging is a complex and progressive biological process that leads to vulnerability …” or similar instead of “Aging is a complex and progressive biological process[1]. It represents a major risk factor 31 for vulnerability”.

Row 34, I suggest to write “impact” or similar instead of “crisis”.

Row 35-36, I suggest to write “Even though several promising anti-aging active molecules have been identified …” or similar instead of “Even though several researches reported promising anti-aging active molecules …”.

Row 37, I suggest to write “ …could enhance quality of life and elongate lifespan …” or similar instead of “….could elongate lifespan….”.

Row 45, I suggest to write “In the TORC1 signalling ..” or similar instead of “In TORC1 signaling …”.

Row 46, I suggest to write “…TORC1 positively regulated…” or similar instead of “…TORC1 and positively regulated…”.

Row 47, I suggest to write “….oxidative stress conditions…” or similar instead of “…oxidative condition…”.

Row 49, I suggest to write “…in the TORC1/Sch9 signal transduction pathway.” or similar instead of “…in TORC1/Sch9 signal transduction.”.

Row 50, I suggest to write “…by the TORC1…” or similar instead of “…by TORC1….”.

Row 53, I suggest to write “…  in the aging process.…” or similar instead of “… in aging process. ...”.

Row 58, I suggest to write “… rDNA circle formations …” or similar instead of “… rDNA circles formations ...”.

Row 69-70, I suggest to write “… Therefore, this method was also used in the current study to screen for …” or similar instead of “… Therefore, we used the same method to screen...”.

Row 73, I suggest to write “… root is used for treatment of…” or similar instead of “… root of this plant is used to treat...”.

Row 76, I suggest to write “… using mixtures …” or similar instead of “… of mixtures ...”.

Row 78, I suggest to write “… existed …” or similar instead of “… existed ...”.

Row 82, I suggest to write “… further studied …” or similar instead of “… taken deep research ...”.

Row 83, I suggest to write “… we identified another …” or similar instead of “… we found another ...”.

Row 86, I suggest to write “… extended lifespan of yeast via inhibition of the TORC1/Sch9/Rim15/Msn signaling…” or similar instead of “… extend the lifespan of yeast via inhibition of TORC1/Sch9/Rim15/Msn signaling ...”.

Row 90, I suggest to write “… columns …” or similar instead of “… column ...”.

Row 91, I suggest to write “… plates …” or similar instead of “… plate ...”.

Row 112, I suggest to write “… an active …” or similar instead of “… active ...”.

Row 112-113, I suggest to write “… through comparison of the spectroscopic results with those previously reported in literature …” or similar instead of “… by comparison of spectroscopic results with reported literature ...”.

Row 129, I suggest to write “… Afterwards …” or similar instead of “… Afterward ...”.

Row 137-138, I suggest to write “… After cultivation for 25 hrs, the cells were inoculated …” or similar instead of “… After the cells were cultured in SC medium for 24 hrs, it was inoculated ...”.

Row 146, I suggest to write “… death …” or similar instead of “… deaths ...”.

Row 162, I suggest to write “… Initially …” or similar instead of “… At first ...”.

Row 164, I suggest to write “… in a shaker at 180 rpm and 28 …” or similar instead of “… in shaker with 180 rpm at 28...”.

Row 170, I suggest to write “… dark. Thereafter,  …” or similar instead of “… dark and then,

...”.

Row 176, I suggest to write “Thereafter, cultivated cells …” or similar instead of “Consequently, the cultured cells ...”.

Row 183, I suggest to write “… with an initial …” or similar instead of “… with initial ...”.

Row 185, I suggest to write “Thereafter, …” or similar instead of “Afterwards,  ...”.

Row 188, I suggest to write “… incubation for …” or similar instead of “… incubating ...”.

Row 197, I suggest to write “Then …” or similar instead of “Afterwards...”.

Row 198, I suggest to write “… cultured cells to get final concentration 10 μM. The cells …” or similar instead of “… cultured cell to get final concentration 10 μM. Cells ...”.

Row 204-205, I suggest to write “The cells were cultured in the same way for ROS determination, they were cleaned with…” or similar instead of “After the cells were cultured in the same way for ROS determination, it was cleaned with…”.

Row 210, I suggest to write “… as previously described …” or similar instead of “… as previous study ...”.

Row 211, I suggest to write “… and shaking at 180 rpm.” or similar instead of “… in 180 rpm shaker.”.

Row 214, I suggest to write “… cells were cultivated for 22 hrs and washed …” or similar instead of “… cultured cells at 22 hrs later, were washed ...”.

Row 220, I suggest to write “… streaked out … one of the colonies…” or similar instead of “… smeared ... one of yeast colony…”.

Row 222, I suggest to write “… the cultures were …” or similar instead of “… the yeasts were  ...”.

Row 242, I suggest to write “… used …” or similar instead of “… empoyed ...”.

Row 243, I suggest to write “… molecules …” or similar instead of “… molecule ...”.

Row 243, I suggest to write “… as controls …” or similar instead of “… trustworthiness  ...”.

Row 251, I suggest to write “… significantly increased as shown in …” or similar instead of “… significantly enhanced  ...”.

Row 267, I suggest to write “Based on this hypothesis…” or similar instead of “Based on this idea...”.

Row 269, I suggest to write “… were investigated …” or similar instead of “… were measured  ...”.

Row 272, Please complete the sentence “given in ...”.

Row 274, I suggest to write “… significant increased as compared to the control group.  …” or similar instead of “… significant increased than that of control group.  ...”.

Row 280, I suggest to write “… yeast cells were…” or similar instead of “… yeast cell was...”.

Row 286, I suggest to write “… gene expression and protein levels …” or similar instead of “… expression, protein level  ...”.

Row 290, I suggest to write “… changes …” or similar instead of “… the changes  ...”.

Row 292, I suggest to write “… antibodies …” or similar instead of “… antibody  ...”.

Row 298, I suggest to write “… treatment with GTS B…” or similar instead of “… treatment of GTS B  ...”.

Row 299, I suggest to write “… protein levels of BY4741 after treatments with GTS B ….“ instead of “… protein level of BY4741 after treating GTS B  ...”.

Row 306, I suggest to write “… or gene expression for these genes …” or similar instead of “… of these genes expression  ...”.

Row 311, I suggest to write “… after treatment with GTS B…” or similar instead of “… treating GTS B  ...”, two places.

Row 312, I suggest to write “… Rim15 is involved in …” or similar instead of “… Rim15 involved ...”.

Row 318, I suggest to write “… treatment with…” or similar instead of “… giving  ...”.

Row 343, I suggest to write “Gene expression …” or similar instead of “The genes expression   ...”.

Row 344, I suggest to write “… after treatment with …” or similar instead of “… after treating with ...”.

Row 351, I suggest to write “… after treatment with …” or similar instead of “… after treating with ...”.

Row 362-364, I suggest to write “… ROS serves as a signalling molecule for physiological responses, causes oxidative modifications to macromolecules like proteins, nucleic acids and lipids at excessive levels.  …” or similar instead of “… ROS which serves as a signaling molecule at physiologic level, causes oxidative modifications to macromolecules like protein, nucleic acid and li-pid at excessive level. ...”.

Row 366, I suggest to write “… potential role as an anti-aging molecule …” or similar instead of “… role in the anti-aging effect ...”.

Row 379, I suggest to write “… were significantly decreased …” or similar instead of “… are significantly decreased ...”.

Row 381, I suggest to write “… was also observed …” or similar instead of “… is also observed ...”.

Row 383, I suggest to write “… trough protection from …” or similar instead of “… via ...”.

Row 394, I suggest to write “… the Replicative Lifespan  …” or similar instead of “… Replicative Lifespan ...”.

Row 403, I suggest to write “… prolonged lifespan …” or similar instead of “… prolong lifespan ...”.

Row 404, I suggest to write “… the effect …” or similar instead of “… effect ...”.

Row 408, I suggest to write “… a replicative lifespan …” or similar instead of “… replicative lifespan ...”.

Row 410, 412, 414, 416, 417 and 422, I suggest to write “… mutants …” or similar instead of “… mutant ...”.

Row 423, I suggest to write “… GTS B related anti-aging effects. …” or similar instead of “… GTS B anti-aging effect. ...”.

Row 455, I suggest to write “… relationship studies and a promising lead compound was through this study identified and developed…” or similar instead of “… relationship study and a promising lead compound was achieved ...”.

Row 457, Delete “in this study”.

Row 458, I suggest to write “… effects …” or similar instead of “… effect ...”.

Row 459, I suggest to write “… The TOR signalling pathway plays …” or similar instead of “… TOR signaling pathway takes ...”.

Row 31-31, I suggest to write “…  …” or similar instead of “…  ...”.

Row 459 and 474, I suggest to write “… the TOR …” or similar instead of “… TOR ...”.

Row 474, I suggest to write “…is involved …” or similar instead of “… involved ...”.

Row 475, I suggest to write “… effects …” or similar instead of “… effect ...”.

Row 476, delete “which”

Row 476-477, I suggest to write “… important anti-oxidative genes …” or similar instead of “… anti-oxidative important genes ...”.

Row 477, I suggest to write “… the Sod and Cat …” or similar instead of “… Sod and Cat ...”.

Row 478-479, I suggest to write “… We focused on further elucidation of gene expression and enzyme activity for selected enzymes related to anti-oxidative stress …” or similar instead of “… We focused on the gene expression and activity of these enzymes to do investigation. we measured the genes expression and enzymes activity related to anti-oxidative stress ...”.

Row 480, I suggest to write “… gene expression …” or similar instead of “… genes expression ...”.

Row 481-482, I suggest to write “… the activity for these enzymes  …” or similar instead of “… these enzymes activity ...”.

Row 484, I suggest to write “… had anti-oxidative stress effects by increasing the gene expression and enzymatic activity for these genes…” or similar instead of “… existed anti-oxidative stress effect via increasing anti-oxidative genes expression and these enzymes activity. ...”.

Row 485, I suggest to write “… To get further elucidate which of these proteins …” or similar instead of “… To get direct evidences which these proteins ...”.

Row 486, I suggest to write “… mutants …” or similar instead of “… the mutants ...”.

Row 487, I suggest to write “… assays. We detected no …” or similar instead of “… assay. The no changes  ...”.

Row 488, I suggest to write “… indicating that these proteins play an important role in anti-aging…” or similar instead of “… indicated that these proteins took important roles in anti-aging …”.

Row 490, I suggest to write “… is sufficient …” or similar instead of “… sufficient …”.

Row 491-492, I am not clear what you want to say with the sentence here. I suggest to write “… formation of the Atg1/Ulk1 complex thereby inhibiting autophagy. …” or similar instead of “… Atg1/Ulk1 complex was prepared to inhibit autophagy. …”. However, check so that it has the meaning you aim for.

Row 498-499, I suggest to write “… indication that the Sir2 signalling pathway is one …” or similar instead of “… inspiration, Sir2 signal pathway is one …”.

Row 500, I suggest to write “… effects …” or similar instead of “… effect …”.

Row 501, I suggest to write “… through …” or similar instead of “… via …”.

Author Response

Reply for reviewer 2

Referee 2

( ) English language and style

( ) English very difficult to understand/incomprehensible
(x) Extensive editing of English language and style required
( ) Moderate English changes required
( ) English language and style are fine/minor spell check required
( ) I don't feel qualified to judge about the English language and style

Yes

Can be improved

Must be improved

Not applicable

Does the introduction provide sufficient background and include all relevant references?

(x)

( )

( )

( )

Are all the cited references relevant to the research?

(x)

( )

( )

( )

Is the research design appropriate?

(x)

( )

( )

( )

Are the methods adequately described?

(x)

( )

( )

( )

Are the results clearly presented?

( )

(x)

( )

( )

Are the conclusions supported by the results?

(x)

( )

( )

( )

Comments and Suggestions for Authors

The authors are presenting their research regarding Gentirigeoside B from G. rigescens and the effect of the compound in yeast related to Modification of lifespan, anti-oxidative stress and autophagy. The article is well presented; however, I have some critical points that I feel needs to be corrected before this manuscript can be accepted for publication.

Between different species, processes, molecular pathways, and homologous genes are not always the same. This problem is not often discussed, especially within pharma applications, and I feel that we need to address this matter more. It is good that animal models are not used in the current study, since animal models are out-dated and should to a larger extent be replaced by more modern way to study pathways and efficacy. I also think that using yeast as a model organism is good since it enables a rapid, cost efficient and sustainable way further explore pathways and mechanisms. However, I would also like to see a small section where a discussion regarding the use of drawbacks when using model organisms, such as yeast that is used in the current study, to solve a problem on another organism, human. I would like the authors to address this topic and to which degree human cell models could have been/can be applied instead.

Below are my specific comments and suggestions for improvement.

Question 1. Throughout the manuscript, please correct the line breaks so that they do not occur in the middle of words. For abbreviations, always write out the full name the initial time that the abbreviation is used, e.g. for BSA at row 147.

Answer 1: We did not find BSA at L147.

Question 2: Row 14, I suggest to write “In the present study evaluation…” instead of “In the present study. The evaluation…”.

Answer 2: We revised “In the present study The evaluation…”  into “In the present study, the evaluation…” in P1, L14.

Question 3: Row 18-19, I suggest to write “were observed after treatment with GTS B. Also, autophagy of yeast ….” if one aspiration timepoint was used or “were observed after treatment of GTS B. At the same time, autophagy of yeast ….”  

Answer 3: We revised “were observed after treatment of GTS B. At the same time, autophagy of yeast ….” into “were observed after treatment with GTS B. Also, autophagy of yeast was also increased” in P1, L18-19.

Question 4: Row 20, I suggest to write “….and reduced the levels of ROS and MDA.” or similar instead of “and reduced ROS and MDA levels in yeast, respectively.”.

Answer 4: We revised “and reduced ROS and MDA levels in yeast, respectively.” into “as well as reduced the levels of ROS and MDA” in P1, L20.

Question 5: Row 21, I suggest to write “It also increased the gene expression and enzymatic…...” or similar instead of “Meanwhile, it increases the genes expression and enzymatic….”.

Answer 5: We revised “Meanwhile, it increases the genes expression and enzymatic….” into “It also increased the genes expression and enzymatic…” in P1, L20.

Question 6: Row 31-32, I suggest to write “Aging is a complex and progressive biological process that leads to vulnerability …” or similar instead of “Aging is a complex and progressive biological process[1]. It represents a major risk factor for vulnerability”.

Answer 6: We revised the sentence into “Aging is a complex and progressive biological process that leads to vulnerability” in P1, L31.

Question 7: Row 34, I suggest to write “impact” or similar instead of “crisis”.

Answer 7: We revised “the burden of aging and age-related crises on socio-economic status…” into “the burden of aging and age-related impact on socio-economic status…” in P1, L34.

Question 8: Row 35-36, I suggest to write “Even though several promising anti-aging active molecules have been identified …” or similar instead of “Even though several researches reported promising anti-aging active molecules …”.

Answer 8: We revised “Even though several researches reported promising anti-aging active molecules …” into “Even though several promising anti-aging active molecules have been identified” in P1, L34-35.

Question 9: Row 37, I suggest to write “…could enhance quality of life and elongate lifespan …” or similar instead of “…could elongate lifespan….”.

Answer 9: We revised “…could elongate lifespan…” into “…could enhance quality of life and elongate lifespan…” in P1, L37.

Question 10: Row 45, I suggest to write “In the TORC1 signalling...” or similar instead of “In TORC1 signaling …”.

Answer 10: We revised “In TORC1 signaling …” into “In the TORC1 signaling pathway…” in P2, L45.

Question 11: Row 46, I suggest to write “…TORC1 positively regulated…” or similar instead of “…TORC1 and positively regulated…”.

Answer 11: We revised “…TORC1 and positively regulated…” into “…TORC1 positively regulated…” in P2, L46.

Question 12: Row 47, I suggest to write “… oxidative stress conditions…” or similar instead of “…oxidative condition…”.

Answer 12: We revised “…oxidative condition…” into “…oxidative stress condition…” in P2, L47.

Question 13: Row 49, I suggest to write “…in the TORC1/Sch9 signal transduction pathway.” or similar instead of “…in TORC1/Sch9 signal transduction.”.

Answer 13: We revised “…in TORC1/Sch9 signal transduction pathway.” into “in the TORC1/Sch9 signal transduction.” in P2, L49.

Question 14: Row 50, I suggest to write “…by the TORC1…” or similar instead of “…by TORC1….”.

Answer 14: We revised “…by TORC1…” into “by the TORC1 signaling pathway…” in P2, L50.

Question 15: Row 53, I suggest to write “…  in the aging process.…” or similar instead of “… in aging process. ...”.

Answer 15: We revised “…in aging process…” into “…in the aging process…” in P2, L53.

Question 16: Row 58, I suggest to write “… rDNA circle formations …” or similar instead of “… rDNA circles formations ...”.

Answer 16: We revised “…rDNA circles formations.” into “…rDNA circle formations.” in P2, L59.

Question 17: Row 69-70, I suggest to write “… Therefore, this method was also used in the current study to screen for …” or similar instead of “… Therefore, we used the same method to screen...”.

Answer 17: We revised “Therefore, we used the same method to screen...” into “…Therefore, this method was also used in the current study to screen for …” in P2, L72-73.

Question 18: Row 73, I suggest to write “… root is used for treatment of…” or similar instead of “… root of this plant is used to treat...”.

Answer 18: We keep the original writing in P2, L76.

Question 19: Row 76, I suggest to write “… using mixtures …” or similar instead of “… of mixtures ...”.

Answer 19: We accepted your suggestion and revised “…of mixtures ...” into “…the memory improving effect using mixtures of gentisides was evaluated…” in P2, L79-80.

Question 20: Row 78, I suggest to write “… existed …” or similar instead of “… existed ...”.

Answer 20: There is no question

Question 21: Row 82, I suggest to write “… further studied …” or similar instead of “… taken deep research ...”.

Answer 21: We revised “…taken deep research ...” into “…mechanisms of action of these molecules were investigated…” in P2, L85.

Question 22: Row 83, I suggest to write “… we identified another …” or similar instead of “… we found another ...”.

Answer 22: We revised “…we found another ...” into “…we identified another …” in P2, L86.

Question 23: Row 86, I suggest to write “… extended lifespan of yeast via inhibition of the TORC1/Sch9/Rim15/Msn signaling…” or similar instead of “… extend the lifespan of yeast via inhibition of TORC1/Sch9/Rim15/Msn signaling ...”.

Answer 23: We revised “…extend the lifespan of yeast via inhibition of TORC1/Sch9/Rim15/Msn signaling ...” into “… extended lifespan of yeast via inhibition of the TORC1/Sch9/Rim15/Msn signaling…” in P2, L89-90.

Question 24: Row 90, I suggest to write “… columns …” or similar instead of “… column ...”.

Answer 24: We revised “…reversed phase C18 open column ...” into “…reversed phase C18 open columns …” in P3, L94.

Question 25: Row 91, I suggest to write “… plates …” or similar instead of “… plate ...”.

Answer 25: We revised “Pre-coated silica gel (0.25 mm) and RP-18 plate…” into “Pre-coated silica gel (0.25 mm) and RP-18 plates…” in P3, L96.

Question 26: Row 112, I suggest to write “… an active …” or similar instead of “… active ...”.

Answer 26: We revised “…to obtain active sample (4.4 mg, 17 minutes) …” into “…to obtain an active sample (4.4 mg, 17 minutes) …” in P3, L116.

Question 27: Row 112-113, I suggest to write “… through comparison of the spectroscopic results with those previously reported in literature …” or similar instead of “… by comparison of spectroscopic results with reported literature ...”.

Answer 27: We revised “…by comparison of spectroscopic results with reported literature ...” into “…through comparison of the spectroscopic results with those previously reported in literature.” in P3, L117-118.

Question 28: Row 129, I suggest to write “… Afterwards …” or similar instead of “… Afterward ...”.

Answer 28: We revised “Afterward, 1 mL of broth…” into “Afterwards, 1 mL of broth…” in P3, L131.

Question 29: Row 137-138, I suggest to write “… After cultivation for 25 hrs, the cells were inoculated …” or similar instead of “… After the cells were cultured in SC medium for 24 hrs, it was inoculated ...”.

Answer 29: We revised “…After the cells were cultured in SC medium for 24 hrs, it was inoculated ...” into “After incubation for 24 hrs, the cells were inoculated…” in P3, L143-144.

Question 30: Row 146, I suggest to write “… death …” or similar instead of “… deaths ...”.

Answer 30: We revised “…as hundred percent deaths.” into “…as hundred percent death.” in P4, L152.

Question 31: Row 162, I suggest to write “… Initially …” or similar instead of “… At first ...”.

Answer 31: We revised “At first, these yeast strains stored in -30…” into “Initially, these yeast strains stored in -30…” in P4, L168.

Question 32: Row 164, I suggest to write “… in a shaker at 180 rpm and 28 …” or similar instead of “… in shaker with 180 rpm at 28...”.

Answer 32: We revised “…in shaker with 180 rpm at 28...” into “…in shaker at 180 rpm and 28…” in P4, L170-171.

Question 33: Row 170, I suggest to write “… dark. Thereafter,” or similar instead of “… dark and then...”.

Answer 33: We revised “… dark and then...” into “…7 minutes in dark. Thereafter” in P4, L176.

Question 34: Row 176, I suggest to write “Thereafter, cultivated cells …” or similar instead of “Consequently, the cultured cells ...”.

Answer 34: We revised “Consequently, the cultured cells ...” into “Thereafter, the cultivated cells…” in P4, L182.

Question 35: Row 183, I suggest to write “… with an initial …” or similar instead of “… with initial ...”.

Answer 35: We revised “…with initial...” into “…with an initial ...” in P4, L189-190.

Question 36: Row 185, I suggest to write “Thereafter, …” or similar instead of “Afterwards,..”.

Answer 36: We revised “…Afterwards...” into “…Thereafter, ” in P4, L191.

Question 37: Row 188, I suggest to write “… incubation for …” or similar instead of “… incubating ...”.

Answer 37: We revised “…incubating ...” into “…incubation for three days.” in P4, L194.

Question 38: Row 197, I suggest to write “Then …” or similar instead of “Afterwards...”.

Answer 38: We revised “…Afterwards...” into “…Then…” in P5, L203.

Question 39: Row 198, I suggest to write “… cultured cells to get final concentration 10 μM. The cells …” or similar instead of “… cultured cell to get final concentration 10 μM. Cells ...”.

Answer 39: We revised “…cultured cell to get final concentration 10 μM. Cells...” into “…cultured cells to get final concentration 10 μM. The cells …” in P5, L204.

Question 40: Row 204-205, I suggest to write “The cells were cultured in the same way for ROS determination, they were cleaned with…” or similar instead of “After the cells were cultured in the same way for ROS determination, it was cleaned with…”.

Answer 40: We revised “After the cells were cultured in the same way for ROS determination, it was cleaned with…” into “The cells were cultured in the same way for ROS determination, they were cleaned with…” in P5, L210-211.

Question 41: Row 210, I suggest to write “… as previously described …” or similar instead of “… as previous study ...”.

 Answer 41: We revised “…as previous study ...” into “…as previously described…” in P5, L215-216.

Question 42: Row 211, I suggest to write “… and shaking at 180 rpm.” or similar instead of “… in 180 rpm shaker.”.

Answer 42: We revised “…in 180 rpm shaker” into “…and shaking at 180 rpm.” in P5, L217.

Question 43: Row 214, I suggest to write “… cells were cultivated for 22 hrs and washed …” or similar instead of “… cultured cells at 22 hrs later, were washed ...”.

Answer 43: We revised “…cultured cells at 22 hrs later, were washed” into “Cells were cultured for 22 hrs and washed with PBS” in P5, L220.

Question 44: Row 220, I suggest to write “… streaked out … one of the colonies…” or similar instead of “… smeared ... one of yeast colony…”.

Answer 44: We revised “… smeared ... one of yeast colony…” into “was spread on YPD agar plates. After forming colonies on plates, one of yeast colonies was inoculated …” in P5, L225-226.

Question 45: Row 222, I suggest to write “… the cultures were …” or similar instead of “… the yeasts were..”.

Answer 45: We revised “… the yeasts were    ” into “the yeast cultures were” in P5, L228.

Question 46: Row 242, I suggest to write “… used …” or similar instead of “… empoyed ...”.

Answer 46: We revised “…employed ...” into “… used …” in P6, L248.

Question 47: Row 243, I suggest to write “… molecules …” or similar instead of “… molecule ...”.

Answer 47: We revised “… molecule …” into “… molecules …” in P6, L249.

Question 48: Row 243, I suggest to write “… as controls …” or similar instead of “… trustworthiness.”.

Answer 48: We revised the sentence into “And RES was used as a positive control.” in P6, L249.

Question 49: Row 251, I suggest to write “… significantly increased as shown in …” or similar instead of “… significantly enhanced.”.

Answer 49: We revised “… significantly enhanced” into “   were also significantly increased as shown in Fig. 1c.” in P6, L256-257.

Question 50: Row 267, I suggest to write “Based on this hypothesis…” or similar instead of “Based on this idea...”.

Answer 50: We revised “Based on this idea..” into “ Based on this hypothesis.” in P6, L273.

Question 51: Row 269, I suggest to write “… were investigated …” or similar instead of “… were measured.”.

Answer 51: We revised “… were measured. ” into “ … were investigated ” in P6, L275.

Question 52: Row 272, Please complete the sentence “given in ...”.

Answer 52: We revised the sentence into “we examined whether GTS B affects the gene expression of SIR2 and the results were shown in Fig. 2(d).” in P7, L278-279.

Question 53: Row 274, I suggest to write “… significant increased as compared to the control group.  …” or similar instead of “… significant increased than that of control group.  ...”.

Answer 53: We revised “… significant increased than that of control group.  ... ” into “… significantly increased as compared to the control group.  … ” in P7, L280.

Question 54: Row 280, I suggest to write “… yeast cells were…” or similar instead of “… yeast cell was...”.

Answer 54: We revised “… yeast cell was... ” into “… yeast cells were  ” in P7, L287.

Question 55: Row 286, I suggest to write “… gene expression and protein levels …” or similar instead of “… expression, protein level”.

Answer 55: We revised “… expression, protein level. ” into “gene expression and protein level ” in P7, L293.

Question 56: Row 290, I suggest to write “… changes …” or similar instead of “… the changes”.

Answer 56: We revised “the changes ” into “ changes ” in P7, L296.

Question 57: Row 292, I suggest to write “… antibodies …” or similar instead of “… antibody.”.

Answer 57: We kept the original writing.

Question 58: Row 298, I suggest to write “… treatment with GTS B…” or similar instead of “… treatment of GTS B.”.

Answer 58: We revised “treatment of GTS B” into “treatment with GTS B ” in P8, L305.

Question 59: Row 299, I suggest to write “… protein levels of BY4741 after treatments with GTS B ….“ instead of “… protein level of BY4741 after treating GTS B  ...”.

Answer 59: We revised “… protein level of BY4741 after treating GTS B  ...” into “protein levels of BY4741 after treatments with GTS B …. ” in P8, L306.

Question 60: Row 306, I suggest to write “… or gene expression for these genes …” or similar instead of “… of these genes expression  ...”.

Answer 60: We revised “… of these genes expression  ...” into “…We also detected the changes of gene expression for these genes. ” in P8, L313.

Question 61: Row 311, I suggest to write “… after treatment with GTS B…” or similar instead of “… treating GTS B  ...”, two places.

Answer 61: We revised “…treating GTS B..” into “…after treatment with GTS B ” in P8, L313-317.

Question 62: Row 312, I suggest to write “… Rim15 is involved in …” or similar instead of “… Rim15 involved ...”.

Answer 62: We revised “…Rim15 involved ..” into “… Rim15 is involved in ” in P8, L320.

Question 63: Row 318, I suggest to write “… treatment with…” or similar instead of “… giving  ...”.

Answer 63: We revised “giving” into “treatment with ” in P8, L326.

Question 64: Row 343, I suggest to write “Gene expression …” or similar instead of “The genes expression   ...”.

Answer 64: We revised “…The genes expression   .” into “…Genes expression” in P9, L351.

Question 65: Row 344, I suggest to write “… after treatment with …” or similar instead of “… after treating with ...”.

Answer 65: We revised “after treating with ... .” into “…after treatment with ” in P9, L352.

Question 66: Row 351, I suggest to write “… after treatment with …” or similar instead of “… after treating with ...”.

Answer 66: We revised “after treating with ... .” into “…after treatment with ” in P10, L359.

Question 67: Row 362-364, I suggest to write “… ROS serves as a signalling molecule for physiological responses, causes oxidative modifications to macromolecules like proteins, nucleic acids and lipids at excessive levels.  …” or similar instead of “… ROS which serves as a signaling molecule at physiologic level, causes oxidative modifications to macromolecules like protein, nucleic acid and li-pid at excessive level. ...”.

Answer 67: We revised “ROS which serves as a signaling molecule at physiologic level, causes oxidative modifications to macromolecules like protein, nucleic acid and li-pid at excessive level.” into “ROS serves as a signalling molecule for physiological responses, causes oxidative modifications to macromolecules like proteins, nucleic acids and lipids at excessive levels  ” in P10, L370-372.

Question 68: Row 366, I suggest to write “… potential role as an anti-aging molecule …” or similar instead of “… role in the anti-aging effect ...”.

Answer 68: We revised “… role in the anti-aging effect ...” into “…potential role as an anti-aging molecule ” in P10, L374.

Question 69: Row 379, I suggest to write “… were significantly decreased …” or similar instead of “… are significantly decreased ...”.

Answer 69: We revised “…are significantly decreased ....” into “were significantly decreased ” in P10, L387.

Question 70: Row 381, I suggest to write “… was also observed …” or similar instead of “… is also observed ...”.

Answer 70: We revised “…is also observed .... .” into “was also observed” in P10, L389.

Question 71: Row 383, I suggest to write “… through protection from …” or similar instead of “… via ...”.

Answer 71: We revised “… via ... .” into “through protection from” in P10, L391.

Question 72: Row 394, I suggest to write “… the Replicative Lifespan  …” or similar instead of “… Replicative Lifespan ...”.

Answer 72: We revised “…Replicative Lifespan.” into “ the Replicative Lifespan .” in P11, L402.

Question 73: Row 403, I suggest to write “… prolonged lifespan …” or similar instead of “… prolong lifespan ...”.

Answer 73: We revised “…prolong lifespan  ” into “ prolonged lifespan.” in P11, L411.

Question 74: Row 404, I suggest to write “… the effect …” or similar instead of “… effect ...”.

Answer 74: We revised “…effect  ” into “ the effect ” in P11, L412.

Question 75: Row 408, I suggest to write “… a replicative lifespan …” or similar instead of “… replicative lifespan ...”.

Answer 75: We revised the sentence into “sod1 mutants with replicative lifespans of 7.50 ± 0. 47 generations for control group” in P11, L416.

Question 76: Row 410, 412, 414, 416, 417 and 422, I suggest to write “… mutants …” or similar instead of “… mutant ...”.

Answer 76: Row 410, 412, 414, 416, 417 “mutant” represents a specific mutant, row 422 “mutant” is an adjective form. We didn't make any changes and kept in P11, L416, 418, 420, 422 and P12, L424, 426, 430.

Question 77: Row 423, I suggest to write “… GTS B related anti-aging effects. …” or similar instead of “… GTS B anti-aging effect. ...”.

Answer 77: We revised “…GTS B anti-aging effect. . ” into “ GTS B related anti-aging effects ” in P12, L431.

Question 78: Row 455, I suggest to write “… relationship studies and a promising lead compound was through this study identified and developed…” or similar instead of “… relationship study and a promising lead compound was achieved ...”.

Answer 78: We revised the sentence into “relationship studies and a promising lead compound was identified and developed through this study” in P13, L463-464.

Question 79: Row 457, Delete “in this study”.

Answer 79: We deleted “in this study” in P13, L465.

Question 80: Row 458, I suggest to write “… effects …” or similar instead of “… effect ...”.

Answer 80: We revised “…effect. . ” into “ effects ” in P13, L466.

Question 81: Row 459, I suggest to write “… The TOR signalling pathway plays …” or similar instead of “… TOR signaling pathway takes ...”.

Answer 81: We revised “TOR signaling pathway takes . ” into “ The TOR signalling pathway plays” in P13, L467.

Question 82: Row 31-31, I suggest to write “…  …” or similar instead of “…  ...”.

Answer 82: There is no question.

Question 83: Row 459 and 474, I suggest to write “… the TOR …” or similar instead of “… TOR ...”.

Answer 83: We revised “ TOR  . ” into “ the TOR  ” in P13, L467, 482.

Question 84: Row 474, I suggest to write “…is involved …” or similar instead of “… involved ...”.

Answer 84: We revised “involved . . ” into “ was involved ” in P13, L482-483.

Question 85: Row 475, I suggest to write “… effects …” or similar instead of “… effect ...”.

Answer 85: We revised “ effect ... . ” into “effects ” in P13, L483.

Question 86: Row 476, delete “which”

Answer 86: We deleted “which” in P13, L484.

Question 87: Row 476-477, I suggest to write “… important anti-oxidative genes …” or similar instead of “… anti-oxidative important genes ...”.

Answer 87: We revised “anti-oxidative important genes... . ” into “important anti-oxidative genes” in P13, L484-485.

Question 88: Row 477, I suggest to write “… the Sod and Cat …” or similar instead of “… Sod and Cat ...”.

Answer 88: We revised “ Sod and Cat ...  ” into “ the Sod and Cat ...” in P13, L485.

Question 89: Row 478-479, I suggest to write “… We focused on further elucidation of gene expression and enzyme activity for selected enzymes related to anti-oxidative stress …” or similar instead of “… We focused on the gene expression and activity of these enzymes to do investigation. we measured the genes expression and enzymes activity related to anti-oxidative stress ...”.

Answer 89: We revised the sentence into “We focused on further elucidation of gene expression and enzyme activity for selected enzymes related to anti-oxidative” in P13, L486-487.

Question 90: Row 480, I suggest to write “… gene expression …” or similar instead of “… genes expression ...”.

Answer 90: We revised “ genes expression ....” into “ gene expression.” in P13, L487.

Question 91: Row 481-482, I suggest to write “… the activity for these enzymes…” or similar instead of “… these enzymes activity ...”.

Answer 91: We revised “ these enzymes activity....” into “ the activity for these enzymes ” in P13, L489.

Question 92: Row 484, I suggest to write “… had anti-oxidative stress effects by increasing the gene expression and enzymatic activity for these genes…” or similar instead of “… existed anti-oxidative stress effect via increasing anti-oxidative genes expression and these enzymes activity. ...”.

Answer 92: We revised the sentence into“GTS B decreased oxidative stress by increasing anti-oxidative genes expression and anti-oxidative enzymatic activity”in P13, L491-492.

Question 93: Row 485, I suggest to write “… To get further elucidate which of these proteins …” or similar instead of “… To get direct evidences which these proteins ...”.

Answer 93: We revised the sentence into“To get further elucidation which of these proteins”in P13, L492- P14, L493.

Question 94: Row 486, I suggest to write “… mutants …” or similar instead of “… the mutants ...”.

Answer 94: We deleted “the” in P14, L493.

Question 95: Row 487, I suggest to write “… assays. We detected no …” or similar instead of “… assay. The no changes...”.

Answer 95: We revised the sentence into“No changes of replicative lifespan of these mutants” in P14, L495.

Question 96: Row 488, I suggest to write “… indicating that these proteins play an important role in anti-aging…” or similar instead of “… indicated that these proteins took important roles in anti-aging …”.

Answer 96: We revised the sentence into “ indicating that these proteins play an important role in anti-aging effect of GTS B.” in P14, L495-496.

Question 97: Row 490, I suggest to write “… is sufficient …” or similar instead of “… sufficient …”.

Answer 97: We revised “… sufficient …” into“… is sufficient …”in P14, L497.

Question 98: Row 491-492, I am not clear what you want to say with the sentence here. I suggest to write “… formation of the Atg1/Ulk1 complex thereby inhibiting autophagy. …” or similar instead of “… Atg1/Ulk1 complex was prepared to inhibit autophagy. …”. However, check so that it has the meaning you aim for.

Answer 98: We revised the sentence into “thereby inhibiting the formation of the Atg1/Ulk1 complex to inhibit autophagy.” in P14, L498-499.

Question 99: Row 498-499, I suggest to write “… indication that the Sir2 signalling pathway is one …” or similar instead of “… inspiration, Sir2 signal pathway is one …”.

Answer 99: We revised the sentence into “It gives us an indication that the Sir2 signalling pathway is one of the key research.” in P14, L505-506.

Question 100: Row 500, I suggest to write “… effects …” or similar instead of “… effect …”.

Answer 100: We revised “…effect…” into“…effects …”in P14, L515.

Question 101: Row 501, I suggest to write “… through …” or similar instead of “… via …”.

Answer 101: We revised “…via…” into“…through ……”in P14, L516.

Reviewer 3 Report

The manuscript presents data demonstrating that a triterpenoid glycoside, gentirigeoside B, isolated from dried roots of Gentiana rigescens Franch, prolongs the replicative and chronologic lifespan of Saccharomyces cerevisiae, and this effect is dependent on the inhibition of the TORC1/Sch9/Rim15/Msn signaling pathway and enhancement of autophagy.

The authors use the ingenious model of the yeast (strain K6001) in which daughters are unable to produce daughter cells in a glucose medium, enabling determination of lifespan by assessment of colony formation by mother cells instead of the tedious micromanipulation procedure.

Remarks:

What was the purity of the gentirigeoside B used?

How was the replicative and chronologic lifespan defined? As time/generations corresponding to 50% survival? It should be indicated in the text.

Were protease (and if so, which) inhibitors used during preparation of extracts for electrophoresis and blotting?

The full name of the plant genus should appear in the title.

Line 14: “In the present study.”  Please remove this phrase or connect it with the next sentence.

Lines 16 and 166: “M”, please insert the exact symbol of unit

Lines 37/38: “and thereby useful”, better “and … be useful”

Line 40: “kinas”, should ne “kinase”

Line 52: “through reduction of reactive oxygen species”, better: “through reduction of the level…”; “reduction of ROS” associated with the chemical reduction while in fact not only chemical reduction is used; eg, Sod reaction involves dismutation i.e. both reduction and oxidation of a radical pair

Line 483: “GTS B existed anti-oxidative stress”, unclear

Line 502 and other: the authors seem to use the term “anti-oxidative stress”. While the term “oxidative stress” is obvious and commonly used, “anti-oxidative stress” is unclear and apparently useless. The authors can replace it via other terms.

There are other language problems apart from those indicated above; linguistic check of the manuscript is recommended.

Author Response

Reply for reviewer 3

Referee 3

( ) English language and style

( ) English very difficult to understand/incomprehensible
( ) Extensive editing of English language and style required
(x) Moderate English changes required
( ) English language and style are fine/minor spell check required
( ) I don't feel qualified to judge about the English language and style

Yes

Can be improved

Must be improved

Not applicable

Does the introduction provide sufficient background and include all relevant references?

(x)

( )

( )

( )

Are all the cited references relevant to the research?

(x)

( )

( )

( )

Is the research design appropriate?

(x)

( )

( )

( )

Are the methods adequately described?

( )

(x)

( )

( )

Are the results clearly presented?

(x)

( )

( )

( )

Are the conclusions supported by the results?

(x)

( )

( )

( )

Comments and Suggestions for Authors

The manuscript presents data demonstrating that a triterpenoid glycoside, gentirigeoside B, isolated from dried roots of Gentiana rigescens Franch, prolongs the replicative and chronologic lifespan of Saccharomyces cerevisiae, and this effect is dependent on the inhibition of the TORC1/Sch9/Rim15/Msn signaling pathway and enhancement of autophagy.

The authors use the ingenious model of the yeast (strain K6001) in which daughters are unable to produce daughter cells in a glucose medium, enabling determination of lifespan by assessment of colony formation by mother cells instead of the tedious micromanipulation procedure.

Remarks:

Question 1. What was the purity of the gentirigeoside B used?

Answer: The purity of the compound is determined to be above 95% by NMR spectrum.

Question 2. How was the replicative and chronologic lifespan defined? As time/generations corresponding to 50% survival? It should be indicated in the text.

Answer: The replicative lifespan is generally defined as the number of progenies produced by the cell division of a single yeast cell until death. Chronological lifespan refers to the survival time of yeast cells during undivided and stable periods. We inserted these sentences in P2, L67-69 and P3, L139-140.

Question 3. Was protease (and if so, which) inhibitors used during preparation of extracts for electrophoresis and blotting?

Answer: When we prepared the protein sample, we added the 1% protease inhibitor cocktail I (#CW2200S, CoWin Biotech, Beijiang, China), 1% phosphatase inhibitor cocktail II (#ab201113, Abcam Trading Ltd. Company, Shanghai, China) and 1% phosphatase inhibitor cocktail III (#ab201114, Abcam Trading Ltd. Company, Shanghai, China) into RAPI lysis buffer (#CW2334, CoWin Biotech, Beijiang, China) to prevent the degradation of protein.

To easily understand for reader, we added “in RAPI lysis buffer” in P5, L230

Question 4. The full name of the plant genus should appear in the title.

Answer: We accepted your comment and revised the title as following:

Gentirigeoside B from Gentiana rigescens Franch Prolongs Yeast Lifespan via Inhibition of TORC1/Sch9/Rim15/Msn Signaling Pathway and Modification of Oxidative Stress and Autophagy

Question 5. Line 14: “In the present study.” Please remove this phrase or connect it with the next sentence.

Answer: We revised it in P1, L14-15 as following:

In the present study, the evaluation of anti-aging effect and action mechanism analysis for this compound were conducted.

Question 6. Lines 16 and 166: “M”, please insert the exact symbol of unit

Answer: we revised it in L16, L172 and L359 as following:

GTS B significantly extended the replicative lifespan and chronological lifespan of yeast at doses of 1, 3 and 10 mM.

Subsequently, the MSN2-GFP yeast at 0.1 initial OD600 in each group were treated with GTS B at 0, 1, 3 and 10 mM and rapamycin at 1 mM as positive control for 2 hrs, respectively.

As indicated in Figures 6(d)-(f), the enzymes activity of them were significantly increased after treating with RES at 10 mM and GTS B at doses of 3 and 10 µM (p < 0.01, p < 0.001, p < 0.001; p < 0.001, p < 0.01, p < 0.05; p < 0.01, p < 0.01, p < 0.01), respectively.

Question 7. Lines 37/38: “and thereby useful”, better “and … be useful”

Answer: We revised this sentence in L36-39 as following:

Therefore, studies for discovery of small molecules which could enhance quality of life and elongate lifespan and thereby acting as a protective measure to alleviate age related disorders and furnish healthy lifespan are highly desirable.

Question 8. Line 40: “kinas”, should ne “kinase”

Answer: We revised “kinas” in L40 into “kinase” as following:

The target of rapamycin (TOR) is a nutrient-sensing protein kinase which is evolutionarily conserved in eukaryotic organisms[4].

Question 9. Line 52: “through reduction of reactive oxygen species”, better: “through reduction of the level…”; “reduction of ROS” associated with the chemical reduction while in fact not only chemical reduction is used; eg, Sod reaction involves dismutation i.e. both reduction and oxidation of a radical pair

Answer: We accepted your kind suggestion.

Question 10. Line 483: “GTS B existed anti-oxidative stress”, unclear

Answer: The complete sentence in P13, L491-492 is as following:

The results in Figure 7 revealed that GTS B decreased oxidative stress by increasing anti-oxidative genes expression and anti-oxidative enzymatic activity.

Question 11. Line 502 and other: the authors seem to use the term “anti-oxidative stress”. While the term “oxidative stress” is obvious and commonly used, “anti-oxidative stress” is unclear and apparently useless. The authors can replace it via other terms.

Answer: We accepted comment and revised them into “reduction of oxidative stress” in full text.

Question 12. There are other language problems apart from those indicated above; linguistic check of the manuscript is recommended.

Answer: We carefully checked grammar and spelling mistakes, please see them in revised manuscript.